# Anvil-radiation diurnal interaction: Shortwave radiative-heating destabilization driving the diurnal variation of convective anvil outflow and its modulation on the radiative cancellation

Zhenquan Wang[1], *

[1] School of Atmospheric Sciences, Nanjing University, Nanjing, China

*Correspondence to:* Zhenquan Wang (zhqwang@smail.nju.edu.cn)

**Abstract.** The behavior of convection producing the anvil is neither well derived from current available observations nor well represented in models. In this work, a novel convective cloud data product is designed to capture the convective anvil outflow. Convective organizations and life stages are derived from the images of infrared brightness temperature (BT) of geostationary (GEO) satellites based on a variable-BT segment tracking algorithm, which brings the possibility for quantifying the convective anvil outflow. Vertical structures of convection are measured by sensors of the A-Train Constellation, which provides the cross section of convective outflow. Here, GEO-based convective tracking and A-Train-detected cloud vertical profiles are combined to develop a novel comprehensive GEO-A-Train merged (GATM) convective cloud data product for investigating the process of convective anvil outflow.

On the basis of this novel Lagrangian-view GATM data, the anvil production for mesoscale convective systems (MCSs) can be quantified. The results show that daytime MCSs can produce more anvil clouds than nighttime MCSs. During the daytime, shortwave radiative heating strongly destabilizes the MCS top to promote the anvil outflow with strong divergence, whereas the nighttime divergence driven by the longwave radiative cooling through radiative destabilization and circulation is weak. Moreover, the assessed sensitivity of the radiative budget to the diurnal-cycle phase shift of the convective anvil outflow is approximately $-1$ W m$^{-2}$ hr$^{-1}$ when the phase shift is in the range between $-4$ and $8$ hr (otherwise the sensitivity has the same magnitude but positive). Stronger diurnal-cycle amplitude can further amplify this sensitivity. Overall, this work presents the observed anvil-radiation diurnal interaction process: radiative heating determines the diurnal variation of anvil outflow; in turn, the diurnal variation of anvil outflow determines the Earth radiative budget.

## 1. Introduction

Tropical convective regions are usually characterized by abundant convective activities and anvil clouds (Houze, 2004; Yuan and Houze, 2010). Anvil clouds have strong interactions with radiation. Nevertheless, the longwave (LW) and shortwave (SW) cloud radiative effects (CREs) of tropical anvil clouds are both large to cancel each other, namely radiative near cancellation (Kiehl, 1994). The final net radiation of tropical convective regions is nearly the same as that of non-convective regions (Hartmann and Berry, 2017). If the radiative cancellation is disturbed, the radiative impacts of anvil clouds on the Earth would be easily amplified to form strong feedback process with the climate, which is the leading uncertainty in the climate sensitivity simulated by climate models (Bretherton, 2015; Hartmann, 2016; Sherwood et al., 2020). The possibility of some not yet understood feedback processes that result in the radiative near cancellation cannot be ruled out (Hartmann, 2016; Sokol et al., 2024; Mckim et al., 2024; Raghuraman et al., 2024).

The radiative near cancellation is by no means guaranteed in the future climate change (Gettelman and Sherwood, 2016). The determinant of the radiative cancellation is important for modulating the Earth radiative budget, and is the bridge of the anvil interacting with the climate. Two theories have been proposed for explaining the radiative near cancellation, but no definite full answer has been provided until now (Hartmann, 2016). Kiehl (1994) argued that the major determinant of the

near cancellation is the cloud-top height that has a weak dependence on the sea surface temperature (SST), and thus the near cancellation across warm and cold oceans is a fortuitous coincidence. As the climate warms, the cloud-top height will rise to enhance the cancellation between LW and SW CREs, which allows the anvil to trap more outgoing LW radiation to form a positive feedback process with the climate (Zelinka and Hartmann, 2010).

In addition, Hartmann and Berry (2017) argued that the radiative near cancellation is caused by the offset between the negative CREs of rainy cores and the positive CREs of the non-precipitating anvil clouds. Since the anvil production of convection depends strongly on radiative heating profiles, the radiative near cancellation is basically constrained by the cloud radiative heating (Hartmann and Berry, 2017). As the climate warms, the changes in the atmospheric state can influence the properties of the anvil produced by tropical convection. The variations in the anvil area and the proportion of thin clouds relative to thick clouds both can alter the radiative cancellation (Berry and Mace, 2014). Bony et al. (2016) suggested that the enhanced upper-tropospheric stability can reduce the convective outflow and anvil cloud fraction. As a result, if the anvil opacity stays the same, this reduction of anvil areas is expected to weaken the radiative cancellation to impose negative feedback on the climate. However, although the tropical high cloud area has a reduction as the climate warms, Sokol et al. (2024) suggested that the high cloud opacity is not fixed but the reduction of thick cloud area is stronger compared with thin clouds. This opacity climate response leads to a higher proportion of thin clouds relative to thick clouds and thus results in a positive climate feedback process (Sokol et al., 2024; Raghuraman et al., 2024).

These two hypotheses provide basic physical understandings for the anvil altitude and coverage climate feedback mechanisms, respectively. However, there is still a lack of consideration of the radiative cancellation caused by the diurnal variation of convective anvil outflow. At daytime, only optically thin cirrus clouds have net warming CREs. However, no matter what the thickness is, nocturnal clouds always have net warming effects on the Earth. These nocturnal clouds can help to compensate the daytime net cooling effects to promote the radiative cancellation. As a result, the diurnal variation of convective anvil outflow determines the degree of radiative cancellation fundamentally.

Although the theory of the radiative cancellation modulated by the diurnal variation of convective anvil outflow is rather simple, the diurnal variation of convective anvil outflow has not been well investigated before. Owing to insufficient sub-grid-scale and microphysical processes in climate models, the parameterized process of convective outflow is dependent on the parameter setting and not deemed trustworthy (Clement and Soden, 2005; Suzuki et al., 2013; Zhao, 2014; Sherwood et al., 2020), and the diurnal cycle of clouds in climate models has significant biases (Nowicki and Merchant, 2004; Yin and Porporato, 2017; Chen et al., 2022; Zhao et al., 2023). Due to the uncertainty in the sub-grid turbulence and microphysical processes of cloud-resolving models, the convective anvil outflow still cannot be well simulated (Matsui et al., 2009; Powell et al., 2012; Zeng et al., 2013; Bretherton, 2015; Atlas et al., 2024). In observations, the organized convective structures and the links between convection and anvil clouds are poorly resolved and the convective life-cycle information is not provided in widely-used Eulerian gridded data, such as the International Satellite Cloud Climatology Project (ISCCP) cloud data product (Rossow and Schiffer, 1991) and the Clouds and the Earth's Radiant Energy System (CERES) project (Minnis et al., 2011; Doelling et al., 2016). In a Lagrangian view, many convections of different life stages are usually clustered and their outflowing anvil clouds are merged in complex convective organizations (CCOs) (Yuan and Houze, 2010; Yuan et al., 2011; Wang and Yuan, 2024). Traditional tracking algorithms poorly distinguish the process of convective anvil outflow, since those anvil clouds that are contributed by many different convections are mixed in traditional fixed-threshold tracking (Wang and Yuan, 2024).

On the basis of hourly infrared brightness temperature (BT) images of geostationary satellites (GEOs), a novel adaptive variable-BT segment tracking algorithm has been proposed to partition the CCO into single-cold-core structures for tracking separately (Wang and Yuan, 2024). Non-precipitating anvil clouds are explicitly associated with unique cold cores. As a result, an advantage of this novel tracking algorithm is to quantify the anvil outflow for the duration of convection in

CCOs. GEO observations and the novel variable-BT segment tracking algorithm provide a foundation to capture the convective anvil outflow. Nevertheless, due to the limitation of passive sensors, the convective vertical structures are not well detected by the GEO radiometer imager. Active sensors of A-Train Constellation, such as Cloud-Aerosol Lidar and Infrared Pathfinder Satellite Observations (CALIPSO) Cloud-Aerosol Lidar with Orthogonal Polarization (CALIOP) and CloudSat Cloud Profiling Radar (CPR), can detect the cloud vertical structures (Stephens et al., 2002). But, the A-Train satellite orbit is sun-synchronous and only observations around 01:30 and 13:30 local time (LT) are available, which are too sparse to track storms or to provide a full picture of convective organizations. It seems that the advantages of GEO and A-Train satellite observations are complementary for describing convective outflow processes in 4 dimensions of space and time.

In this work, on the basis of the novel variable-BT segment tracking algorithm, the GEO-based convective tracking and cloud vertical profiles from A-Train satellites are combined to develop a novel comprehensive 4-D GEO-A-Train merged (GATM) convective cloud data product for convective anvil outflow. With this novel 4-D GATM convective cloud data product, the anvil-radiation interaction is systematically investigated on the diurnal time scale for two interactive processes:

**(1) How does the radiation influence diurnal variation of convective anvil outflow?**

**(2) How does the diurnal variation of convective anvil outflow influence the radiative cancellation?**

This work is laid out as follows: Section 2 introduces the data and methods; Section 3 shows the diurnal variation of convective anvil outflow and its modulation by radiative heating, to answer the first question; Section 4 evaluates the sensitivity of the radiative cancellation to the diurnal variation of convective outflow, to answer the second question. Section 5 presents conclusions.

## 2. Data and methods

Section 2 introduces data and methods. Section 2.1 introduces the convective tracking algorithm and data set, which were developed in Wang and Yuan (2024). Section 2.2 introduces the A-Train constellation and the CALIPSO-CloudSat-CERES-MODIS (CCCM) data set, which were developed in Kato et al. (2011). Section 2.3 introduces a novel 4-D GATM convective cloud data product by combining GEO-based convective tracking and A-Train satellite-detected cloud vertical profiles, which is developed in this work. Section 2.4 introduces the study domain that this work focuses on. Section 2.5 introduces statistical methods used in this work.

### 2.1 GEO-based adaptive variable-BT segment tracking data set

Tropical convection usually has complicated organizations and behaviors. It has long been observed that most of mesoscale convective systems (MCSs) are not isolated convective bodies but many MCSs are connected and clustered in a large CCO (Yuan and Houze, 2010; Yuan et al., 2011; Yuan and Houze, 2013). Those MCSs that are connected in CCOs could be initiated at different times and be in different life stages, but their produced anvil clouds are mixed in CCOs. On the basis of hourly GEO BT images at 10.8 μm, Wang and Yuan (2024) have developed a novel variable-BT segment tracking algorithm to partition the CCO into organization segments (OSs) of single cold cores and then to track OSs separately. Here, these OSs can also be understood as convective activities with core structures in CCOs. This algorithm can be used to track the structural evolution of OSs and the anvil production for their durations.

On the basis of hourly GEO images, the steps of the variable-BT segment tracking algorithm are briefly introduced as follows:

**(1) Identification and segmentation of OSs in CCOs:** A set of adaptively variable BT thresholds from 180-260 K per 5-K interval and a minimum area threshold of 1000 km$^2$ are used to identify the CCO structures, i.e., cold cores (the local coldest BT isotherm) and cold centers (the warmest BT isotherm of enclosing only one cold core). These identified core

structures are used to distinguish different convective activities clustered in CCOs. For their segmentation, the pixels lying outside the centers are assigned to the centers iteratively by the 1-K interval on the basis of the nearest route distance (Figure 2 in Wang and Yuan (2024)), which requires that the outflowing anvil clouds must be connected with its origin of cold cores. In this way, anvil clouds in CCOs are explicitly associated with unique cold cores. And the OS is a well-organized single-core structure in 3 dimensions (x, y and BT), in which the cold-core BT can represent its developing strength.

**(2)** **Tracking OSs via dynamic overlaps:** Dynamic overlaps combine the cross correlation and area overlaps for tracking, and refer to the overlap in areas after moving OSs to the position predicted by cross correlation. For two OSs with sufficient dynamic overlap in their major core structures exceeding 50%, they are deemed the same convection at different times. The life cycle of a convective activity clustered in CCOs consists of these temporally associated OSs. The convective peaking strength is represented by the cold-core-peak BT, which is defined as the coldest cold-core BT in life cycles. The life cycle is separated into two stages of the development and decay by the cold-core peaking time of the coldest BT with the largest core area.

In this tracking dataset, the spatial resolution is 0.05° and the temporal resolution is 1 hour. For precipitation, the hourly fine-scale global precipitation measurement (GPM) is collocated with the GEO BT images to provide estimates of pixel-level precipitation (Huffman, 2023). In the tropics, light precipitation (<1 mm/hour) is difficult to be accurately identified by the GEO-based precipitation estimate (Tian et al., 2009) and contributes to only 9%–18% of the total precipitation (Yuan and Houze, 2010). Thus, the threshold of 1 mm/hour is used to distinguish precipitating/non-precipitating region, which is consistent with Yuan and Houze (2010). The MCSs are commonly identified as the tropical deep cloud systems with heavy precipitation (Williams and Houze, 1987; Fu et al., 1990; Yuan and Houze, 2010). Here, the heavy precipitation event is defined as that the area of the precipitation larger than 6 mm/hour exceeds 1000 km$^2$, which is consistent with the definition in Yuan and Houze (2010). The MCS is defined as the OS of the heavy precipitation, the cold-core-peak BT colder than 220 K and the duration over 5 hours. These MCSs represent the cold and long-lived OSs in CCOs and contribute to most of tropical precipitation and anvil clouds (Wang and Yuan, 2024).

For the all-sky radiative flux at the top of atmosphere (TOA), the hourly broadband shortwave albedo ($\beta$) and outgoing longwave radiative flux (LWRF) images of 0.05° resolution are derived from the GEO radiometers. For the clear-sky radiative flux at the TOA, the hourly insolation and the broadband clear-sky LWRF and reflected shortwave radiative flux (SWRF) at each grid of 0.05° are allocated from the CERES synoptic 1-degree (SYN1deg) product (Doelling et al., 2016). The CREs are defined as the difference of the predicted TOA clear-sky upwelling radiative flux relative to the observed TOA all-sky radiative flux:

$$LW\ CRE = LWRF_{clr} - LWRF_{obs}, \tag{1}$$

$$SW\ CRE = SWRF_{clr} - \beta_{obs} \times Insolation_{obs}, \tag{2}$$

$$Net\ CRE = LWCRE + SWCRE, \tag{3}$$

where the subscripts "clr" and "obs" represent the clear sky and the observed all sky, respectively. The CRE represents the TOA radiative energy budget altered by clouds per square meter and per hour. Thus, the impact of the MCS on the radiative energy budget depends largely on the CRE, area and duration. For example, short-lived and small MCSs may have strong CREs but contribute to only a limited energy disturbance, since they only impact a small region during a short time. The CREs of long-lived and large MCSs may not be strong but the radiative energy budget can be strongly altered by them, since they are long-lasting to impact a large region. As a result, to evaluate the impact of the MCS on the radiative energy budget, the radiative energy contribution (REC) is defined as sum of CRE for non-precipitating anvils over their entire area and lifetime:

$$REC = \sum_{i=1}^{D} \sum_{j=1}^{N} CRE_{i,j} \times \delta area_{i,j} \times \delta t. \tag{4}$$

Here, $\delta area$ is the observational grid area of 0.05° resolution, which is a function of latitude. $\delta t$ is the observational time interval, which is 1 hour in this work. The subscript "$i$" represents the $i$-th time and $D$ is the duration of the MCS. The subscript

"*j*" represents the *j*-th grid of the non-precipitating anvil and $N$ is the total number of grids covered by non-precipitating anvil clouds.

Wang and Yuan (2024) have validated this GEO-based tracking by comparing the tracked OS motion with the radiosonde-observed cloud-top winds. The mean speed difference between them is −1.6m/s and the mean angle difference is 0.5°. The motion of cloud systems and the cloud-top winds are not expected to be exactly the same. The magnitude of this bias is well acceptable compared with previous studies (Santek et al., 2019; Daniels et al., 2020).

Although the BT threshold of 260 K is useful for identifying tropical high clouds (Minnis et al., 2008; Minnis et al., 2011), much the area of detrained thin cirrus of the BT warmer than 260 K is not well identified (Gasparini et al., 2022; Sokol and Hartmann, 2020; Berry and Mace, 2014). It has been demonstrated that 95% of deep convective clouds and as much of the anvil cloud as possible can be identified with the least contamination from lower-level clouds by using the threshold of 260 K (Yuan and Houze, 2010; Yuan et al., 2011; Chen and Houze, 1997).

## 2.2 A-Train Constellation

The A-Train Constellation includes five satellites at the altitude of 705 km, which were placed in a tight formation to move in the matched sun-synchronous orbit (Stephens et al., 2002; L'ecuyer and Jiang, 2010). More than a dozen instruments were carried by the satellites in the A-Train. By taking advantage of the tight flying formation of the A-Train, instruments on board different satellites in the A-Train can be combined to simultaneously measure the atmospheric humidity and radiation.

In the A-Train Constellation, CALIOP operates at the wavelength of 1024 nm and 532 nm and is sensitive to small particles but easily attenuated for thick clouds (Winker et al., 2009; Winker et al., 2010). 95-GHz CPR can penetrate thick clouds but usually miss thin cirrus clouds and the upper portion of deep convective clouds, owing to its low sensitivity to small ice crystals (Dessler et al., 2006; Berry and Mace, 2014). The combination of the CALIOP and CPR can provide more accurate full cloud profiles than using either of them individually. The Moderate Resolution Imaging Spectroradiometer (MODIS) is a 36-channel whiskbroom scanning radiometer and can be used to derive cloud optical properties (Minnis et al., 2011). The CERES instrument measures broadband TOA radiances (Wielicki et al., 1996).

In the product of the CCCM developed in Kato et al. (2011), the capabilities of these instruments are merged to contribute to comprehensive cloud-radiation-interaction observations. The vertical cloud profile information from the combined CALIOP and CPR (three 333-m resolution CALIOP profiles and one 1.4-km CPR profile) are matched to the MODIS-derived cloud properties with the horizontal resolution of 1 km (Kato et al., 2010). The combined cloud observations are further collocated with the CERES radiance measurements of the resolution of 20 km (Kato et al., 2011). The cloud fraction at each height is computed as the percentage of clouds detected by CALIOP and CPR over the CERES footprint.

In this CCCM product, LW and SW irradiance vertical profiles were computed via the Fu-Liou radiative transfer model (Fu and Liou, 1993; Fu et al., 1997; Ham et al., 2017; Ham et al., 2022), by inputting the cloud information of cloud top and base pressure from CALIOP and CPR and water phase, particle size, and optical depth from MODIS (Kato et al., 2011). The vertical resolution of cloud fraction and irradiance profiles is 240 m. The radiative heating rate (Q) is computed as:

$$Q = \frac{R_a \cdot T}{p \cdot c_p} \cdot \frac{[F^{\downarrow}(z_2) - F^{\downarrow}(z_1)] - [F^{\uparrow}(z_2) - F^{\uparrow}(z_1)]}{z_2 - z_1}, \tag{5}$$

where $R_a$ is the specific gas constant of dry air, $T$ is temperature, $p$ is pressure, and $c_p$ is the specific heat capacity for dry air at constant pressure. $F^{\downarrow}(z_2)$ and $F^{\downarrow}(z_1)$ are downward irradiance at heights of $z_2$ and $z_1$, respectively. $F^{\uparrow}(z_2)$ and $F^{\uparrow}(z_1)$ are upward irradiance at heights of $z_2$ and $z_1$, respectively. Atmospheric temperature and pressure profiles in the CCCM are from the Goddard Earth Observing System (GEOS-5) Data Assimilation System reanalysis (Kato et al., 2011). LW, SW and net radiative heating rates are computed via Eq. (5) by imputing the LW, SW and net upward and downward irradiance estimates

from the CCCM product. Owing to the sun-synchronous orbit of the A-Train, these radiance measurements in the CCCM are only accessible at around 01:30 and 13:30 LTs.

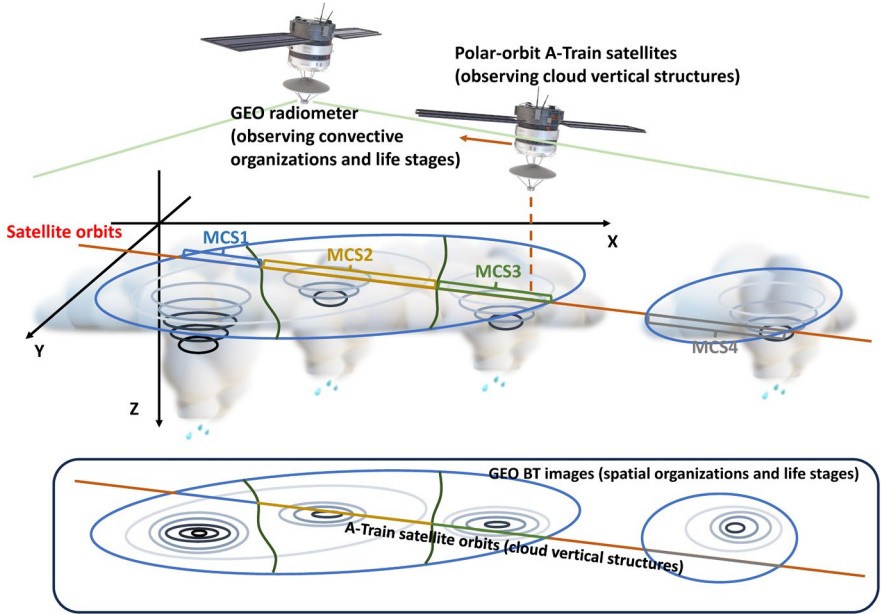

**Figure 1. Schematic diagram of the GATM convective cloud data product.**

**2.3 The GATM convective cloud data product**

The GATM is designed to combine the GEO-based MCS tracking and the A-Train-observed cloud vertical profiles. The GEO-based MCS tracking provides the information of convective organizations and life stages (Section 2.1), while the A-Train satellites provide the information of cloud vertical structures (Section 2.2).

The schematic diagram of the GATM convective cloud data product is presented in Figure 1. The GEO-based MCS 210 tracking and the CCCM are simply collocated by using longitude and latitude. Specifically, the procedures of constructing the GATM are as follows:

**(1) Matching the A-Train-observed profiles with GEO image pixels:** According to the longitude and latitude of each profile in the A-Train orbit and the satellite overpass time, the A-Train profile is matched with the nearest GEO image pixel, with the distance no more than 3 km and the observational time difference no more than half an hour. As illustrated in 215 Figure 1, the orbit of the A-Train corresponds to four GEO-observed MCSs (MCS1-3 are connected and MCS4 is isolated). By matching the A-Train orbit with the GEO image pixels, the association of these A-Train profiles with the GEO-observed MCSs is distinguished.

**(2) Deriving the organization and lifecycle information from the GEO for the A-Train-observed profile:** The organization and lifecycle information for the A-Train and GEO matched profiles in the first step is further derived from the 220 GEO-based MCS tracking. As illustrated in Figure 1, for the MCS1, the A-Train-observed cross section consists of multiple MCS1-related profiles. The GEO provides a full picture of the MCS1 spatial organization structure and its life cycle. In this work, for each MCS1-related profile, the information derived from the GEO includes: a) the distance to the cold-core centroid, which corresponds to its relative position in the MCS; b) the cold-core BT and cold-center BT, which represent the MCS organization structure; c) the cold-core-peak BT, its life stages (developing, peaking or decaying) and the time relative to the 225 convective peaking time, which reflect the MCS life cycle. In this way, these A-Train and GEO matched profiles receive a Lagrangian perspective from the GEO-based MCS tracking.

The GATM combines the advantage of the GEO radiometer imagers in tracking and the advantage of the A-Train active sensors in detecting cloud vertical structures and radiance. In essence, the GEO-based adaptive variable-BT segment tracking data set (Section 2.1) and the A-Train CCCM data set (Section 2.2) are collocated in the GATM. For the GEO-A-Train matching process, a limitation might result from the difference in the temporal resolution between the GEO and A-Train satellites. The GEO images have hourly resolution, whereas the A-Train-observed profiles are instantaneous. Although the observational time difference between the GEO and A-Train satellites is constrained by no more than half an hour for matching, clouds could vary significantly during this half an hour. The temporal resolution of the new-generation GEO has achieved 2.5 mins (e.g., Himawari-8 launched in 2014) and benefits tracking in recent years (Daniels et al., 2020; Yang et al., 2023), but the best observational period of the A-Train Constellation is during the year 2006-2011 (Kato et al., 2011). Thus, in this work, to take advantages of the A-Train Constellation, the GEOs of the same period during 2006-2011 are used to construct the GATM.

Overall, the GATM provides a unique perspective to capture the process-level convective anvil outflow in 4 dimensions of space (x, y and z) and time. If only using the CloudSat, the identification of the convective anvil depends on the whether the convective pillar is observed (Igel et al., 2014; Takahashi and Luo, 2012; Deng et al., 2016; Hu et al., 2021). In other words, if the overpass of the CloudSat is not through the core of the MCS, the anvil cannot be determined. As a result, the anvil identification is limited by the satellite orbit. When producing statistics of the convective cloud structure by only using the CloudSat, the core structure of the MCS is composited by more cloudy samples and fewer clear-sky samples, whereas the edge of the MCS away from the core is the composite of fewer cloudy samples and more clear-sky samples. This might lead to a bias in the composited anvil structure further from the core, since it is mixed with a large number of clear-sky samples. On the other hand, by combining the GEO that provides a full picture of the convective system, even though the overpass of the A-Train is not through the core of the MCS, the convective anvil still can be identified (e.g., MCS1 and MCS4 in Figure 1). In this way, for the composites of the anvil structure, none of clear-sky samples is included and thus the composites of the anvil radiative heating structure further from the core is not biased by clear-sky samples.

## 2.4 Study domain

The study region is selected as the Tropical West Pacific (TWP, 130°W-170°E, 20°S-20°N). Convective activities and anvil clouds are common in this region but the net radiation shows small differences compared with nonconvecting regions. Thus, it is a typical region for investigating the oceanic convection and radiation cancellation. The study time is constrained in June, July and August between 2006 and 2011, to avoid the influence of seasonal cycles. For quality control, the tracked life cycles touching the edges or involving missing images are excluded from analyses. Normally, large and long-lived cloud systems are more likely to touch the edge of the domain than small and short-lived systems. Thus, removing the systems that intersect with the edge can result in a low bias in the number of the large and long-lived systems (Dewitt and Garrett, 2024; Dewitt et al., 2024).

## 2.5 Statistical methods

The 95% confidence interval for the mean value was computed via the $t$ test: $\bar{x} \pm t_c \frac{s}{\sqrt{N}}$, where $\bar{x}$ is the mean value of all samples; $t_c$ is the critical value for $t$; and $s$ is the standard deviation of all the samples. N is the number of independent samples, which is determined based on the e-folding length of autocorrelation (Bretherton et al., 1999).

## 3 How does the radiation influence the diurnal variation of convective anvil outflow?

In the novel Lagrangian-view convective cloud data product, anvil clouds in CCOs are explicitly assigned to MCSs, and the vertical cloud and radiative-heating structures of MCSs are provided. The quantification of MCS anvil production via tracking and the cross section of MCS structures detected by A-Train active sensors are accessible. In this section, the basic features of MCS life cycles are presented in Section 3.1. The diurnal variation of convective anvil outflow and its underlying radiative mechanism are investigated in Section 3.2.

### 3.1 Life cycles of MCSs via adaptive variable-BT segment tracking

The life cycles of MCSs are sorted by the cold-core-peak BT. The BT can be influenced by atmospheric and surface emission and cloud optical properties. Normally, over tropical oceans, the variation of the cold BT (< 230 K) is deemed to be more correlated with the convection activity, whereas the variation of the warm BT could be more influenced by the disturbance in surface emission (Fu et al., 1990; Hendon and Woodberry, 1993). In this work, only the convective peak BT colder than 220 K is considered and the least identified variation of the peak BT is 5 K, which means the fluctuations less than 5 K (that might be contributed by the variation of atmospheric and surface emission) are filtered out. Thus, smaller convective peak BT values are largely contributed by more intense convection. The life cycle is simply separated into two stages: development (before peaking) and decay (peaking and after peaking). The life cycle of convection can be split into more detailed stages, such as the initiation, growth, mature and dissipation (Futyan and Del Genio, 2007; Wall et al., 2018; Yang et al., 2024), or discriminated according to the area variation (Chen and Houze, 1997; Roca et al., 2017). For this work, the convective peak BT is used as a simple useful constraint on the convective intensity to split life cycle stages. Normally, time of the convective peak BT will tend to occur earlier in the life cycle than the largest area (Futyan and Del Genio, 2007; Yang et al., 2024).The decay stage after the time of the convective peak BT (defined in this work) corresponds to the sum of mature and dissipation stages in previous studies (Futyan and Del Genio, 2007; Yang et al., 2024). Wang and Yuan (2024) have validated that the convective peak BT can well constrain the convection precipitating and producing anvil clouds in the life cycle, even though the life cycle is complicated.

Figure 2 shows the basic characteristics of the tracked MCSs over the warm pool of the TWP in June, July and August during 2006-2011. In Figure 2a, during the observation period over the TWP, the MCSs of the peak BT at 205 K have the highest occurrence frequency, with the sample number of 16744. The MCSs of the peak BT at 185 K have the lowest occurrence frequency, with the sample number of only 39. In Figure 2b, for different convective peak BT of MCSs, their relative contributions to anvil areas are computed as the sum of the anvil areas produced by the MCSs in each bin of cold-core-peak BT values in development and decay stages, respectively, divided by the total anvil areas produced by all observed MCSs. Overall, the decay stage of MCSs contributes to 77.5% of total anvil areas, whereas the development stage has relatively small anvil contribution fraction of 22.5%.

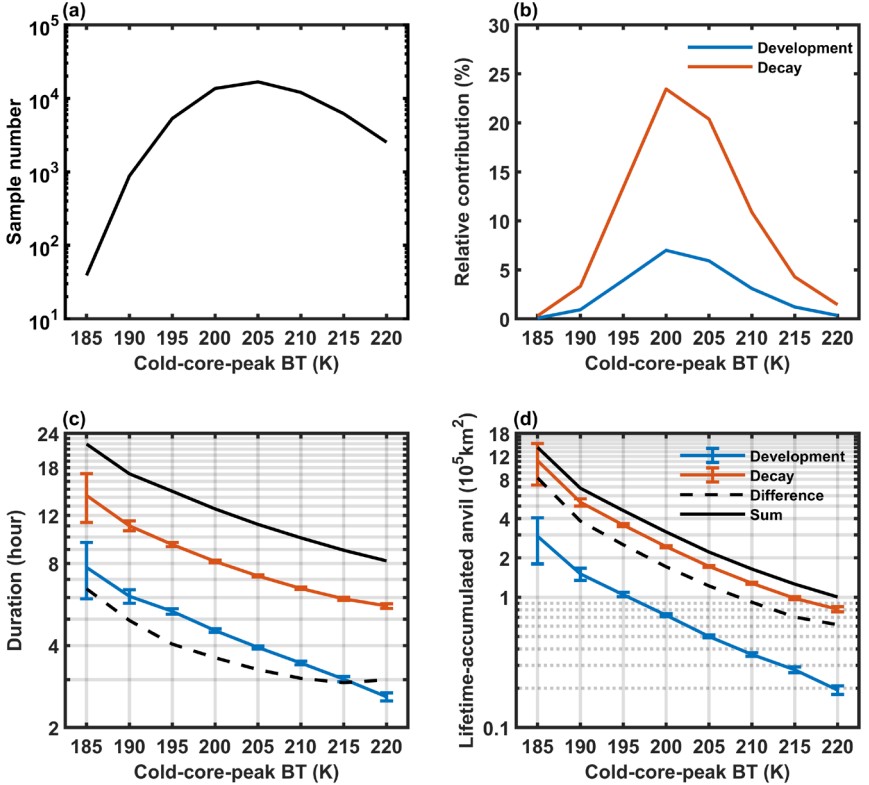

**Figure 2. Life cycles of MCSs via adaptive variable-BT tracking.** (a) The sample number of the tracked MCS life cycles of different cold-core-peak BT in June-August from 2006 to 2011 over the TWP. (b) The relative contribution fractions to total anvil areas for development (the blue line) and decay (the red line) stages, respectively. The average duration (c) and lifetime-accumulated anvil area (d) of different cold-core-peak BT. The blue and red represent the quantities related to the development and decay stages, respectively. The black dash line indicates their differences and the black solid line indicates their sum. The error bars indicate the 95% confidence intervals of the means based on the t test.

Figure 2c and Figure 2d show the average duration and lifetime-accumulated anvil area of MCSs in two stages of development and decay, respectively. Here, the accumulation of anvil areas for the tracked MCS durations is computed as:

$$Accumulated\ anvil\ area = \sum_{i=1}^{D}(A_i \times \delta t), \tag{6}$$

where $A$ represents the anvil area associated with the tracked MCS in each GEO image, and the subscript "$i$" represents the i-th image for the MCS duration. $D$ represents the MCS duration. $\delta t$ corresponds to the observational time interval and in this work is 1 hour. Notably, the Y axes in Figures 2c-d are displayed at the log scale. Thus, the average duration and accumulated anvil area conform to loglinear relationships with the cold-core-peak BT. It implies that the average duration and accumulated anvil area increase exponentially with colder peak BT values, and thereby the MCSs of colder peak BT values are much more long-lived and are accompanied with stronger anvil outflow. On average, the tracked duration for the weakest MCSs of the peak BT at 220 K is 8.2 hours, with the development of 2.6 hours and the decay of 5.6 hours. In contrast, the strongest MCSs of the peak BT at 185 K can persist 21.9 hours on average, in which the development takes 7.7 hours and the decay takes 14.2 hours. For the accumulated anvil area, anvil clouds produced by the strongest MCSs are approximately 14 times those of the weakest MCSs and most of anvil clouds are produced in the MCS decay stage. With MCSs peaking at colder BT values, the difference of duration and accumulated anvil area between the development and decay stages (the black lines in Figures 2c-d) also has an exponential increase roughly. It manifests that the decay process of MCSs corresponds to the main process of convective anvil outflow.

Overall, the accumulated anvil area produced by the MCS has a strong dependence on the convective peak BT, with a log-linear relationship with the cold-core-peak BT. Thus, it is necessary to distinguish the convective strength for discussing

the response of convective anvil outflow to the radiation. In the MCS life cycle, the decay period of MCSs is mainly responsible for the anvil-producing process. For the total anvil cloud budget, the MCSs of the cold-core-peak BT at 200-205 K are the most important, since the anvil area produced by warmer MCSs are small and colder MCSs are not frequent.

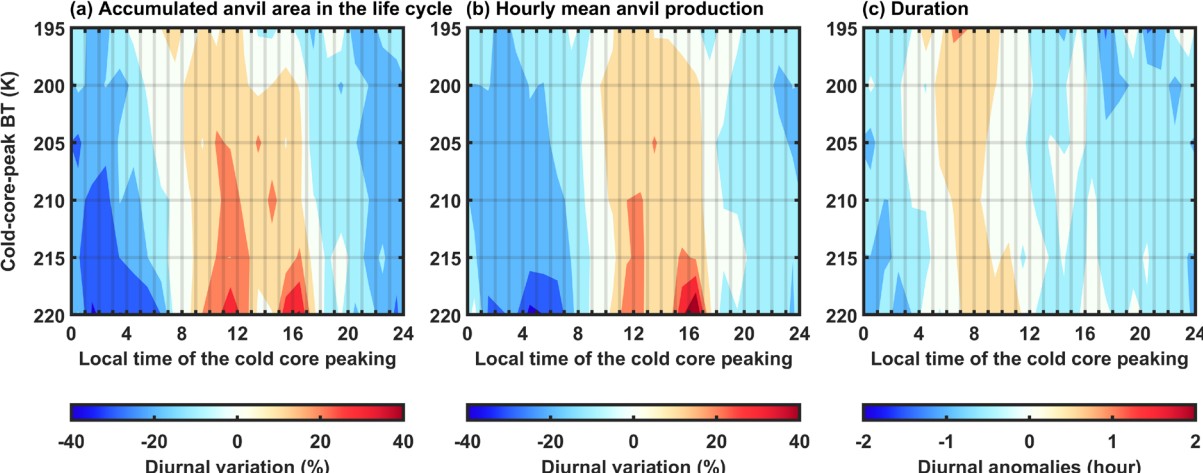

**Figure 3. Diurnal variations of the MCS producing the anvil.** (a-c) Observed diurnal variations of accumulated anvil area produced by MCS, hourly-mean anvil production and duration in the decay stage, respectively.

### 3.2 Observed diurnal variations of convective anvil outflow and its modulation by radiation

To investigate the variability of accumulated anvil area produced by MCSs at the diurnal time scale, the convection strength is constrained by the cold-core-peak BT and the MCSs of different peak BT are discussed separately. For the same convective peak BT, the diurnal variation of the MCSs that peak at different LTs is shown in Figure 3.

Here, only the MCSs of the peak BT from 195-220 K and the decay process are considered. The MCSs of the peak BT at 185-190 K account for only 4.6% of the anvil contribution and have insufficient samples to investigate the variations at the diurnal time scale (Figures 2a-b). The MCSs in each bin of the peak BT from 195-220 K have thousands of samples for investigating their diurnal variations and are the major source of anvil clouds. In addition, the MCS of different LTs will experience very distinct radiative heating profiles at the diurnal time scale, which can significantly influence the development and decay stages both. Here, only the decay process of MCSs is discussed, since most anvil clouds are produced from the decay process. As a result, the diurnal difference in the accumulated anvil area produced by MCSs in Figure 3 can be understood as variations in the MCS decay process accompanied by different radiative heating profiles.

For the accumulated anvil area produced by MCSs, the hourly mean anvil production and duration are two major contributing factors for its diurnal variation. Here, the hourly mean anvil production is defined as the accumulated anvil area divided by the lifetime during the decay stage, which represents the hourly mean anvil area produced in the MCS decay process. In Figure 3, by constraining the cold-core-peak BT, the diurnal variation of the accumulated anvil area, hourly mean anvil production and duration are shown. Here, for the same convective peak BT, the diurnal variation is quantified by the diurnal anomalies at different LTs divided by the mean value. In Figure 3a, the accumulated anvil area produced by MCSs has a significant diurnal cycle. Overall, daytime-peak MCSs can produce more anvil clouds than nocturnal-peak MCSs in their decay processes. In Figure 3b, the hourly anvil production experiences a significant diurnal cycle with the amplitude up to approximately 40% of the average value. But in Figure 3c the amplitude of the MCS duration anomalies is less than one hour. Overall, there is more variation in the average area of anvils than the anvil lifetime with the diurnal cycle.

Similarly, Wall et al. (2020) also reported that the daytime convection can produce more anvil clouds based on wind

tracking. They inferred that the diurnal increase in the accumulated anvil area results from the prolonged lifetime or larger spread areas of anvil clouds. It is confirmed that the convection of the peaking time between 06:00 and 12:00 LT can persist a longer lifetime as shown in Figure 3c, but the diurnal-cycle amplitude of the MCS duration is relatively small. Larger spread areas of anvil clouds during daytime seems to be the main reason for the diurnal variation in the accumulated anvil area produced by MCSs.

Radiative heating is the fundamental explanation for the diurnal variation of convection. Radiation provides the main impetus for large-scale circulation adjustments on the diurnal time scale to result in the diurnal variation of convection (Ruppert and Hohenegger, 2018). Two mechanisms have been proposed to explain the modulation of the circulation and convective anvil outflow by radiative-heating vertical and horizonal gradients, respectively:

(1) **The lapse-rate mechanism (vertical):** The vertical radiative-heating destabilization of the cloud layer stimulates the in-cloud convection and promotes microphysical recycling, thus increasing the convective anvil outflow (Lilly, 1988; Hartmann et al., 2018).

(2) **The differential radiation mechanism (horizontal):** The horizontal gradient of the radiative heating between anvil clouds and the surrounding clear sky invigorates the upper-level circulation, thus increasing the convective anvil outflow (Gray and Jacobson, 1977; Nicholls, 2015; Wall et al., 2020);

These two mechanisms are both reasonable. But, validating them requires quantification of convective anvil outflow, which is neither well parameterized in models nor provided in the previous observational cloud data product.

On the basis of the novel GATM convective cloud data product, all relevant measurements of radiance and atmospheric states are provided for the daytime-peak (13:30 LT) and nocturnal-peak (01:30 LT) MCSs, whose diurnal variation is shown in Figure 3. These measurements can be further used to calculate the structure of convective anvil outflow in MCSs according to the thermodynamic energy equation at steady state and continuity function (Thompson et al., 2017):

$$\vec{V} \cdot \nabla_h T - \omega S = Q, \tag{7}$$

$$D = -\frac{\partial \omega}{\partial p} = \frac{\partial}{\partial p}\left(\frac{Q}{S}\right) + \frac{\partial}{\partial p}\left(\frac{-\vec{V} \cdot \nabla_h T}{S}\right). \tag{8}$$

$\vec{V} \cdot \nabla_h T$ is the horizontal temperature advection, $\omega$ is the vertical velocity in pressure coordinates, $S$ is the stability ($-\frac{T}{\theta}\frac{\partial \theta}{\partial p}$), Q is the diabatic heating, and $D$ is the divergence. In tropics, with weak horizontal gradient of temperature (Sobel et al., 2001), the divergence is largely caused by the diabatic heating and can be simplified as $D = \frac{\partial}{\partial p}\left(\frac{Q}{S}\right)$. As a result, the divergence directly caused by the radiative heating to the MCSs corresponds to the lapse-rate mechanism, whereas the circulation invigorated by the difference of radiative heating between MCSs and the surrounding clear-sky environment corresponds to the differential radiation mechanism. The impacts of these two mechanisms on the diurnal variation of convective anvil outflow are discussed separately as follows.

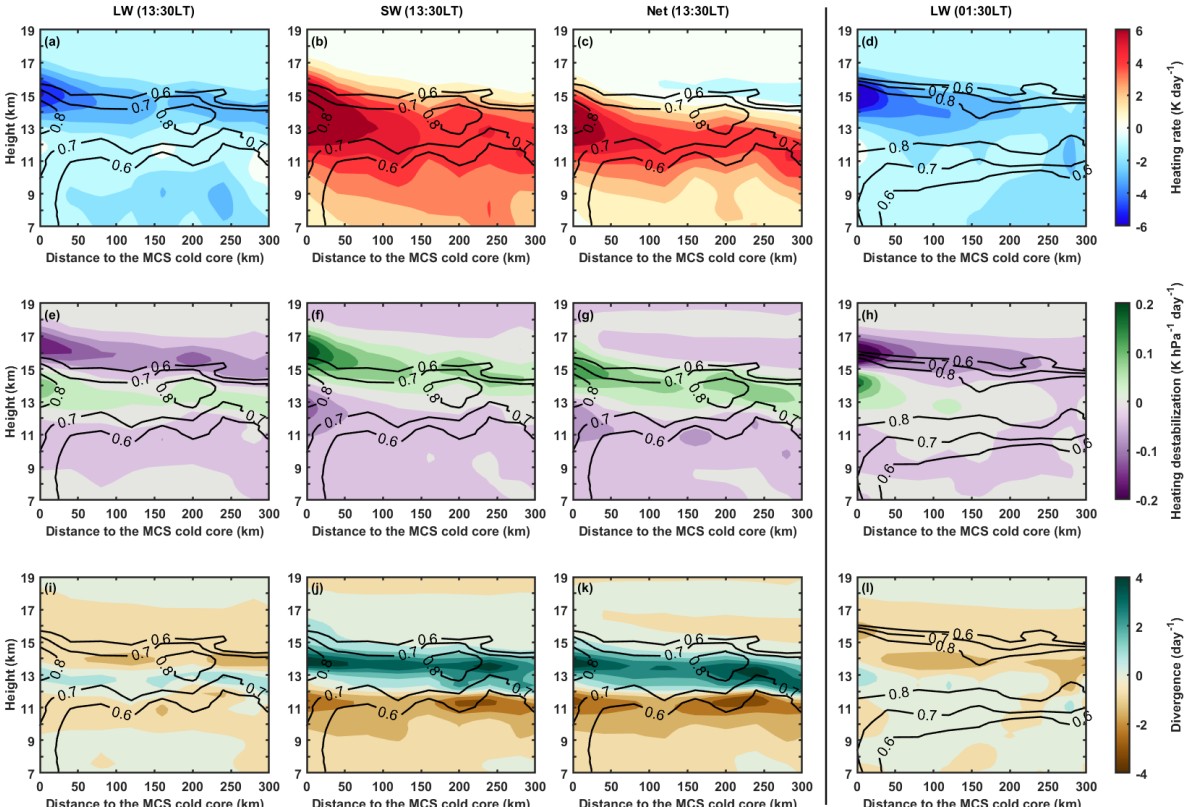

Figure 4. Radiative heating rate, heating destabilization and divergence caused by LW, SW and net radiances for MCSs at 13:30 and 01:30 LT, respectively. (a-c) The LW, SW and net heating rate at 13:30 LT. (d) The LW heating rate at 01:30 LT. (e-h) and (i-l) similar to (a-d) but for heating destabilization and divergence, respectively. The black contours represent cloud fraction.

***For the lapse-rate mechanism***, the heating rate ($Q$, Figures 4a-d), heating destabilization ($\frac{\partial Q}{\partial p}$, Figures 4e-h) and radiation-driven divergence ($\frac{\partial}{\partial p}\left(\frac{Q}{s}\right)$, Figures 4i-l) caused by LW, SW and net radiances are investigated for the MCSs of the cold-core-peak at 13:30 LT (the left panels in Figure 4) and 01:30 LT (the right panels in Figure 4), respectively. The MCS cloud structure is represented by the cloud fraction (the black contours in Figure 4), which is defined as the ratio of cloud occurrence at each vertical level to the number of all samples. Notably, only the profiles with anvil clouds are considered (see more details in Section 2.3). The cloud fraction profile shown in Figure 4 should be understood as the mean vertical distribution of clouds, whereas the cloud fraction profiles shown in previous studies represent the cloud incidence relative to all cloudy and clear-sky samples (Igel et al., 2014). Thus, cloud fraction shown in Figure 4 represents the main structure of cloud vertical distribution. For either 13:30 or 01:30 LT, clouds have well organized structures in MCSs. A convective pillar that is shaped by cloud-fraction contours exists within 50 km around the MCS cold core. Away from the convective pillar, clouds concentrate on the layer between approximately 12-14 km, which well shapes the convective anvil outflow.

At 13:30 LT, LW cooling is nearly completely offset by SW heating, and the MCS is experiencing strong net radiative heating, as shown in Figure 4a-c. In the vertical direction, the net heating of the MCS top has a rapid decline from approximately 13 to 16 km (Figure 4c). The vertical decline of the heating can lead to the radiative destabilization (Figure 4g) and thereby strong convective outflow (Figure 4k) at the MCS top. Overall, the convective outflow at 13:30 LT should be largely attributed to the divergence caused by the SW heating destabilization (Figure 4f and 4j), whereas the LW heating stabilizes the MCS top (Figure 4e) and contributes to little or even negative anvil outflow (Figure 4i).

At 01:30 LT, only LW cooling exists (Figure 4d). And the decline of the LW cooling stabilizes the MCS top (Figure 4h) to inhibit the divergence in the upper portion of MCSs (Figure 4l). In summary, daytime SW heating destabilizes the MCS

top to promote anvil outflow, whereas nighttime LW cooling stabilizes the MCS top to reduce anvil outflow. This diurnal difference of radiative destabilization (the lapse-rate mechanism) is well consistent with the diurnal variation of convective outflow presented in Figure 3. It is also interesting to notice that the nighttime MCSs accompanied with weak divergence seem to have thicker anvil clouds in comparison with the daytime MCS structure shaped by cloud-fraction contours, although the anvil thickness is not the focus of this work.

Figure 5 shows the diurnal difference in the average column net heating rates, destabilization and divergence for MCSs (the dashed lines in Figure 5), whose details are presented in Figure 4. Overall, at 13:30 LT, MCSs are strongly heated by net radiance, and the rapid decline of the net heating rate in the upper troposphere (Figure 5a) contributes to strong radiative destabilization (Figure 5c) and divergence (Figure 5e). At 01:30 LT, the MCS top would be strongly cooled by LW radiance, and the decline of the cooling rate with height (Figure 5b) stabilizes the upper portion of MCSs (Figure 5d) and inhibits divergence (Figure 5f). The divergence at 01:30 LT in Figure 5f corresponds to the region of the increasing cooling rate below the MCS top between approximately 11 km and 13 km. The nighttime divergence (Figure 5f) driven by radiative destabilization occurs at a lower level and is much weaker compared with that at 13:30 LT (Figure 5e). This might explain that the anvil clouds produced by nighttime MCSs are fewer (shown in Figure 3) but seem to be thicker with a lower base than those produced by daytime MCSs (shown in Figure 4).

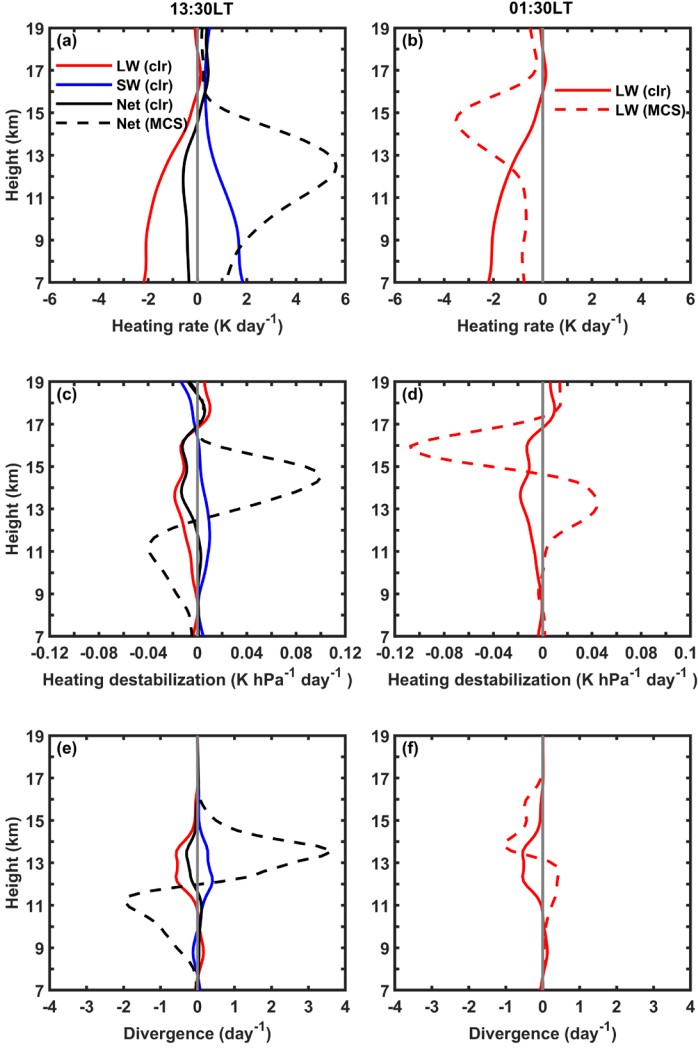

**Figure 5. The composited profiles of radiative heating rate, heating destabilization and divergence at 13:30 and 01:30 LTs, respectively.** (a) The LW, SW and net heating rate at 13:30 LT. (b) The LW heating rate at 01:30 LT. (c-d) and (e-f) similar to (a-b) but for heating destabilization and divergence, respectively. The red, blue and black solid lines represent quantities caused by LW, SW and net radiances for the clear sky, respectively. The black dashed line represents the quantity caused by the net radiance for MCSs, whose details are shown in Figure 4.

***For the differential radiation mechanism***, the circulation between the MCS and its surrounding clear-sky environments results from the balance between the MCS divergence and the clear-sky convergence. Thus, the circulation can be enhanced by either stronger MCS divergence or stronger clear-sky convergence (or vice versa). This provides a perspective that the clear-sky radiative cooling can constrain the convective development through circulation (Hartmann and Larson, 2002). Gray and Jacobson (1977) and Nicholls (2015) argued that the enhancement of the radiative cooling outside the cloud system

can result in stronger clear-sky convergence to promote the circulation, and ultimately leading to the development of convection.

From the perspective of the circulation between the MCS and its surrounding clear sky, the circulation can be split into two parts: cloudy-sky divergence and clear-sky convergence. As discussed above, the divergence at the MCS top is enhanced by SW radiative heating during the daytime (as discussed above and shown in Figure 4), which also represents the

enhancement of the circulation caused by SW heating. Additionally, the circulation also depends on the convergence driven by the clear-sky radiative cooling. In Figure 5, the diurnal variation of the clear-sky convergence induced by radiative cooling is presented. Here, the clear sky refers to no clouds above 7 km based on the CCCM data. For the clear sky, LW radiative cooling at 13:30 LT is partially offset by the SW radiative heating (Figure 5a), whereas only LW cooling is available at 01:30 LT (Figure 5b). Thus, the net cooling at 01:30 LT is stronger than that at 13:30 LT, which can enhance the clear-sky radiative

stabilization and convergence at 01:30 LT as shown in Figure 5c-f. As a result, from the perspective of the circulation, stronger clear-sky convergence at 01:30 LT can strengthen the nighttime convective outflow.

In comparison with the radiative heating and divergence structure of MCSs (the dashed lines in Figure 5), the clear-sky convergence and the MCS divergence are nearly at the same level in Figure 5e-f, which is well consistent with the hypothesized process constrained by mass conservation in Hartmann and Larson (2002). Overall, the lapse-rate and differential

radiation mechanisms both can contribute to the MCS anvil production. In Figure 5e, at 13:30 LT, the MCS divergence directly driven by the radiative destabilization (the rapid decline of the net radiative heating with height) is much stronger than the clear-sky convergence (the differential radiation mechanism). This might imply that daytime MCS anvil production is primarily dominated by the lapse-rate mechanism with a small contribution from the differential radiation mechanism. In Figure 5f, at 01:30 LT, the MCS divergence driven by the radiative destabilization (the increase of the net radiative cooling with height) is

smaller than the clear-sky convergence. This might imply that the nighttime MCS anvil production is largely driven by the clear-sky convergence with a small contribution from the vertical radiative destabilization. At the diurnal time scale, the divergence determined by the radiative destabilization at 13:30 LT is much stronger than the divergence driven by the clear-sky radiative cooling through the circulation at 01:30 LT. As a result, daytime MCSs produce much stronger anvil clouds than nighttime MCSs, which leads to the diurnal variation shown in Figure 3.

**4 How does the diurnal variation of convective anvil outflow influence the radiative cancellation?**

The non-precipitating anvil outflow of MCSs is a key component of the cloud water budget and plays an important role in the Earth's radiative budget (Zhao et al., 2016). But the water and radiative budgets have a non-linear relationship, which means the radiative energy budget cannot be simply inferred from the variation in the anvil amount. A key parameter that determines the non-linear cloud-radiation relationship is the LT of the convection producing the anvil. Different LTs

correspond to various insolation strengths and surface outgoing longwave radiation. Thus, at different LTs, clouds of the same radiative properties (i.e., albedo and longwave radiative emission) can have completely different radiative effects. In this section, on the basis of the GEO-based tracking data set (Section 2.1), the sensitivity of the radiative energy budget to the diurnal variation in the convective anvil production is investigated.

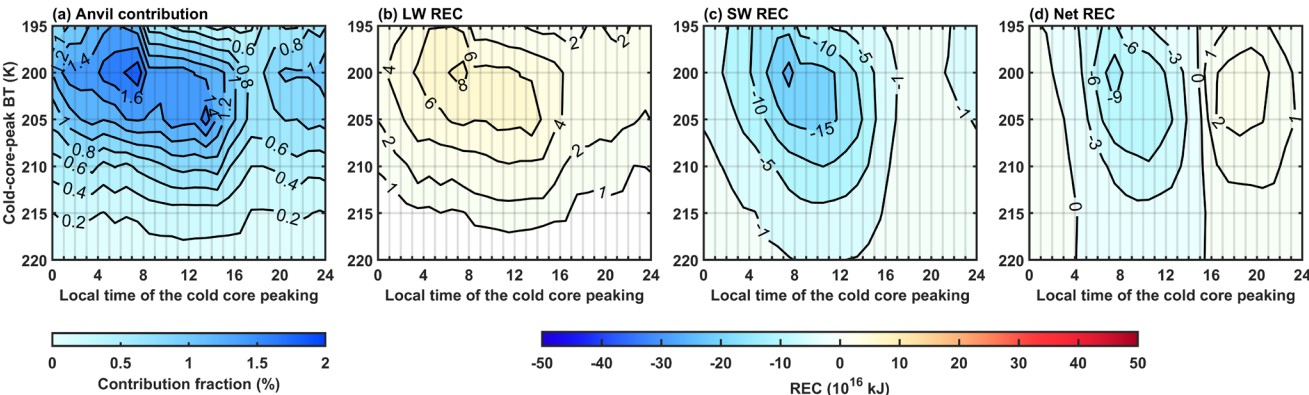

Figure 6. Anvil contribution and radiative cancellation at the diurnal time scale. (a) Anvil contribution fraction of MCSs of different peak BT values and LTs. (b-d) The LW, SW and Net REC caused by MCSs of different peak BT values and LTs.

In Figure 6, anvil clouds and the TOA radiative energy budget contributed by MCSs of different peak BTs and LTs are shown. Only the decay process is considered. The REC represents the changes of the TOA radiative energy budget that result from the non-precipitating anvil clouds produced by MCSs (Eq. 4 and see more details in Section 2.1). The positive sign implies warming effects and the negative sign implies cooling effects on the Earth. The anvil contribution fraction refers to the fraction of the non-precipitating anvil area produced by MCSs of different peak BTs and LTs relative to the anvil area produced by all observed MCSs. Although the radiative energy budget is sensitive to many anvil properties (e.g., the top temperature, structure and life cycle), only the association of the radiative energy budget with anvil area coverage is focused in this section.

Notably, the diurnal variation in the anvil contribution shown in Figure 6a should be attributed to the covariation of the occurrence frequency and the convective anvil outflow of MCSs at the diurnal time scale. As discussed in Section 3, daytime MCSs tend to produce more anvil clouds than the nighttime MCSs. On the other hand, it has been known that the occurrence of deep convection tends to peak at the early morning (Gray and Jacobson, 1977; Chen and Houze, 1997; Nesbitt and Zipser, 2003; Wall et al., 2020). At the diurnal time scale, for the peak BT values at 195-200 K, the MCSs of the peak time at the early morning have a larger anvil contribution than the MCSs that peak at other LTs. For the peak BT value at 205-220 K, the MCSs of the peak LT at the noon or the early afternoon have a larger anvil contribution than that at other LTs. Overall, the MCSs of the peak BT values at 200-205 K have the largest contribution to the anvil area (this is also shown in Figure 2).

For the LW and SW radiative energy budget shown in Figure 6b-c, the LW and SW RECs are closely correlated with the anvil contribution shown in Figure 6a, with the pattern correlations of 0.97 and -0.8 at the 99% significant level, respectively. Notably, although the diurnal-cycle phases of the LW and SW RECs are similar to the anvil contribution, their diurnal-cycle amplitudes are very different, particularly for the SW REC. In Figure 6c, for the peak BT at 200 K, the SW REC of the MCS peaking time at 08:00 LT is the strongest (around -20×10$^{16}$ kJ) and is the smallest for that of the peaking time between 16:00-20:00 LT (around -1×10$^{16}$ kJ), in which the diurnal difference exceeds 20 times. The diurnal-cycle phase of the SW REC is similar to the diurnal cycle of the 200-K MCS anvil contribution, but its diurnal-cycle amplitude is much stronger than that of the anvil contribution. This is simply because the MCSs peaking in the early morning can produce more daytime anvil clouds whose occurrence time corresponds to strong insolation, whereas the MCSs peaking in the late afternoon produce mostly nighttime anvil clouds with little insolation. By coupling with the diurnal cycle of the insolation, the diurnal-cycle amplitude of the SW REC is much stronger than that of the anvil contribution. On the other hand, since the diurnal variation of the surface outgoing LW radiation is not very strong as compared with that of insolation, the diurnal-cycle amplitude of the LW REC and anvil contribution only has a small difference. Consequently, the cancellation ratio between LW (Figure 6b) and SW (Figure 6c) RECs is not constant at the diurnal time scale, but has a strong dependence on the LT of the MCS peak.

As shown in Figure 6d, during 04:00-15:00 LT, the SW REC is only partially cancelled by the LW REC, which leads to a net cooling effect. On the other hand, at other LTs, the SW REC is completely cancelled by the LW REC, with a net

warming effect. Ultimately, a secondary radiation cancellation between the net cooling and warming occurs at the diurnal time scale. 27% of the net cooling are further cancelled by the net warming at the diurnal time scale.

The observed radiation cancellation in Figure 6 might result from many factors. The anvil-top temperature is important for outgoing LW radiation and accounts for the primary radiative cancellation (Kiehl, 1994). Additionally, the negative CREs caused by thick anvil clouds can be partially balanced by the positive CREs of optically thin cirrus clouds and thereby the anvil structure (i.e., the ratio of thin cirrus clouds relative to thick clouds) is also important for the radiative cancellation (Berry and Mace, 2014). Recent studies suggested that the anvil structure is an important determinant of the anvil radiative climate feedbacks (Mckim et al., 2024; Raghuraman et al., 2024; Sokol et al., 2024). Moreover, the diurnal variation of anvil clouds produced by MCSs also can affect the radiative cancellation, particularly for the secondary radiation cancellation at the diurnal time scale. For example, if the diurnal cycle of the MCS anvil contribution (shown in Figure 6a over oceans only) has a positive phase shift, the anvil cloud budget would be redistributed at the diurnal time scale, with relatively more daytime anvil clouds and less nighttime anvil clouds. As a result, the diurnal-cycle amplitude of the SW RECs can be further amplified to reduce the secondary net radiation cancellation ratio and increase net cooling effects. Similarly, the diurnal variations of the anvil-top temperature and anvil structure produced by MCSs are also important for the secondary net radiation cancellation.

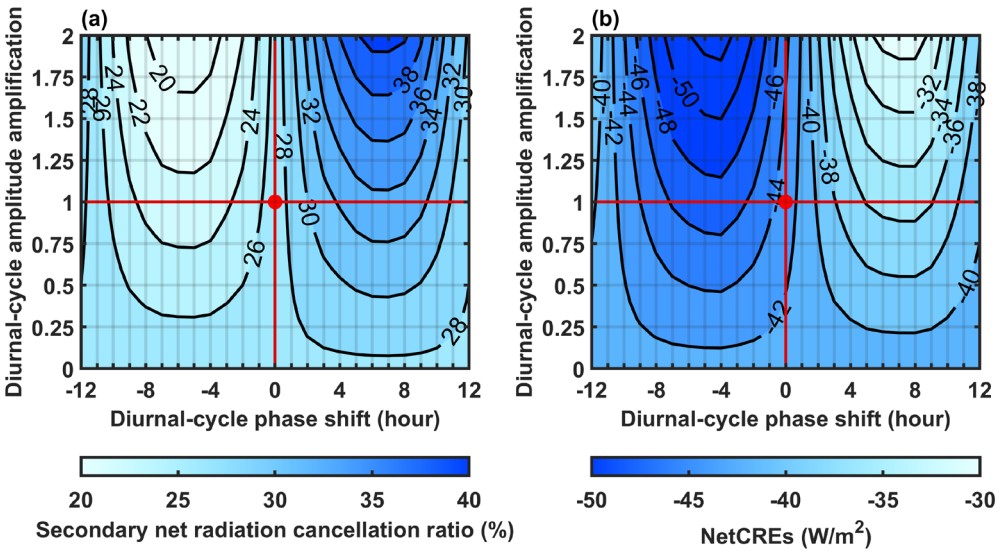

**Figure 7. Sensitivity of the net radiation cancellation to diurnal variations of the MCS producing the anvil.** (a-b) The sensitivity of the secondary net radiation cancellation ratio and net CREs, respectively, to distinct climatology of diurnal variation of MCS anvil production. The red dot represents the current climatology of the diurnal cycle.

Although the diurnal-cycle climate feedback and its relevant mechanism are still missing in current climate studies, cloud-resolving models and observations both suggested that the diurnal cycle of cloud coverage could have a positive phase shift with surface temperature rising (Yin and Porporato, 2019; Gasparini et al., 2021; Wang et al., 2022). Nevertheless, recent studies have suggested that the diurnal-cycle features of clouds (e.g., diurnal-cycle amplitude and phase) simulated by climate models have significant biases compared with the observed realistic diurnal cycle (Nowicki and Merchant, 2004; Yin and Porporato, 2017; Chen et al., 2022; Zhao et al., 2023). As a result, predicting the diurnal-cycle climate feedback remains challenging but might be important for understanding the climate sensitivity. Here, diurnal-cycle climate feedback can be largely determined by multiplying the response of the diurnal cycle to the surface temperature by the sensitivity of the radiative budget to the changes in the diurnal cycle. The response of the diurnal cycle is still an uncertain aspect in the future climate, but the radiative sensitivity to the diurnal cycle can be assessed.

In this work, notably, only the diurnal variation in the convective anvil outflow over oceans (shown in Figure 3 and

discussed in Section 3) is focused and the sensitivity of the radiative energy budget to the diurnal variation in the convective anvil outflow is evaluated in Figure 7. It can also be regarded as the observational constraint of the diurnal variation for model climate projections (Williamson et al., 2021).

In Figure 7, a linear model of the net REC (NetREC) to the MCS anvil production is constructed as:

$$NetREC(BT, \ LT) = F(BT, \ LT)[\alpha A(BT, \ LT) + \beta]. \tag{9}$$

$F$ and $A$ represent the occurrence frequency and the accumulated anvil area produced by MCSs, respectively. NetREC, $F$ and $A$ all are functions of the cold-core-peak LT and BT. $\alpha$ and $\beta$ are the linear regression coefficient and intercept, respectively. For example, $\alpha A(BT, \ LT) + \beta$ in Eq. 9 can be understood as the prediction of the NetREC caused by a MCS of the cold core peak at $BT$ and $LT$ with the accumulated anvil area $A$ for its duration. $F$ corresponds to its occurrence frequency that also has a significant diurnal variation (Gray and Jacobson, 1977; Chen and Houze, 1997; Nesbitt and Zipser, 2003; Wall et al., 2020). Here, only the diurnal variation of $A$ is focused (shown in Figure 3 and discussed in Section 3).

By multiplying the predicted NetREC of $A(BT, \ LT)$ by the MCS occurrence frequency ($F$) in Eq. 9, the final NetREC induced by all anvil production of the MCSs of different BTs and LTs can be computed. The diurnal cycle of $A$ is tunable and can be further expressed as:

$$A(BT, LT) = \overline{A_{obs}(BT)} + \lambda[A_{obs}(BT, LT - \delta) - \overline{A_{obs}(BT)}]. \tag{10}$$

The subscript "obs" represents the value derived from observations. The bar over the letter head represents the mean value. $\lambda$ refers to the diurnal-cycle amplitude amplification ratio. $\delta$ refers to the diurnal-cycle phase shift. For $\lambda = 1$ and $\delta = 0$, $A$ has the diurnal cycle that is consistent with the observation presented in Figure 3a. By altering $\lambda$ and $\delta$, the observed diurnal-cycle amplitudes and phases of $A$ can be tuned. In this way, $A$ with different diurnal cycles of Eq. 10 can be further input to the linear model of Eq. 9 to evaluate the sensitivity of the secondary net radiation cancellation ratio to the diurnal cycle of $A$, as shown in Figure 7a. Here, the secondary net radiation cancellation ratio refers to the cancellation ratio between the net cooling and warming that occurs at the diurnal time scale (27% for the current climatology of diurnal cycles). In Figure 7b, the net CRE (NetCRE) is defined as the $NetREC/(F \times A)$, which can be understood as the sensitivity of the net radiative budget to the anvil-amount variation under different climatology of diurnal cycles.

Figure 7 shows that the net radiation cancellation and NetCREs vary with the changes in the diurnal cycle of convective anvil outflow. If the $A$ diurnal cycle of current climatology (Figure 3) has a positive phase shift, more anvils would be distributed at the nighttime and the secondary cancellation would be enhanced to make the Earth warmer than the current state (or vice versa). If the diurnal-cycle amplitude is invariant ($\lambda = 1$), approximately 10.8% for the secondary cancellation and 11.9 W m$^{-2}$ for NetCREs can be modulated by the diurnal-cycle phase of convective anvil outflow. As a result, for $\lambda = 1$, the sensitivity of NetCREs to the diurnal-cycle phase of convective anvil outflow is approximately -1 W m$^{-2}$ hr$^{-1}$ when the phase shift is in the range between -4 and 8 hr (otherwise the sensitivity has the same magnitude but positive). Notably, the radiative sensitivity to the diurnal-cycle phase is linearly proportional to the diurnal-cycle amplitude amplification ratio ($\lambda$) in Figure 7b. As the diurnal-cycle amplitude is stronger with the amplification ratio $\lambda$, the radiative sensitivity to the phase shift would be amplified by multiplying by $\lambda$. As a result, if the climate response of the diurnal cycle of the convective anvil outflow to the temperature can be known, the sensitivity that is assessed here might be useful for inferring to its feedback strength.

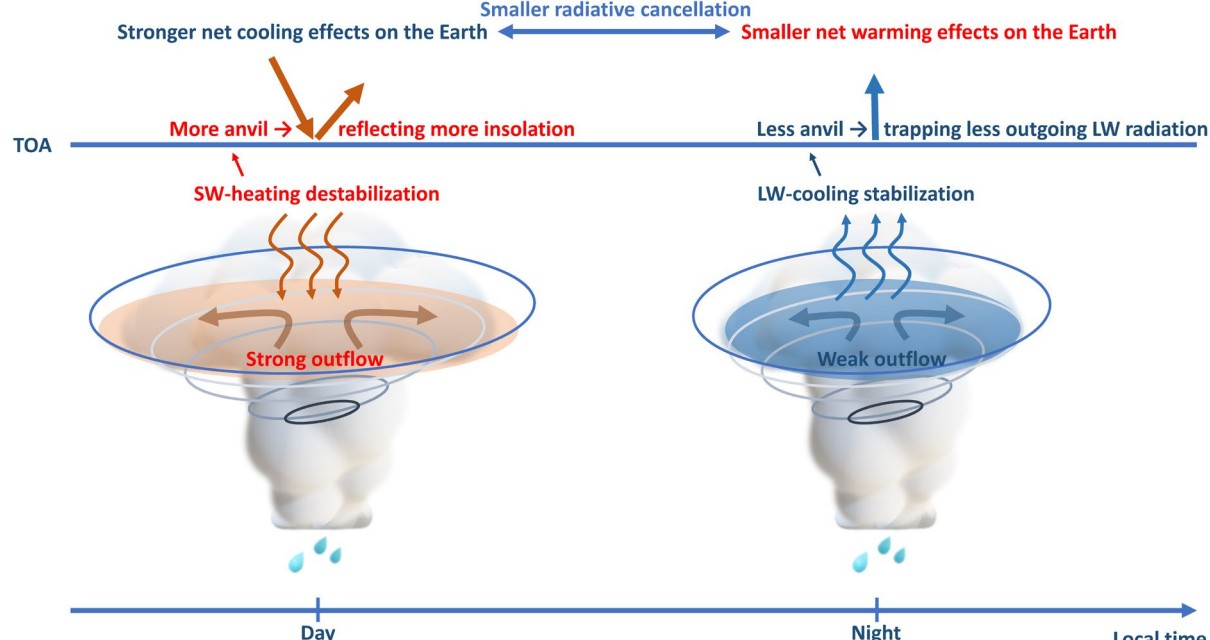

**Figure 8. Illustrations of the anvil-radiation diurnal interaction processes.**

## 5 Conclusions

Tropical convection usually has complex convective organizations. Many convections in different life stages are clustered in complex organizations and their produced anvil clouds are merged. As a result, the process of convective anvil outflow has been poorly distinguished in observations for complex convective organizations.

The observed process of convection producing anvil clouds is the important reference for understanding the convective water budget and developing model convection parameterization. Nevertheless, conventional Eulerian gridded observational product provides limited reference for this process. For example, the widely-used Eulerian gridded data (e.g., ISCCP and CERES project) provide little information of the sub-grid convective organizations and the links between convection and anvil clouds. Although the cloud data product based on the sensors of A-Train Constellation (i.e., CCCM) provides cloud and radiance vertical structures, the full picture of horizonal convective organizations and the life stage of convection are not accessible. In this work, to provide observational reference for convective anvil outflow in complex organizations, the advantages of GEOs and A-Train Constellation are combined with two steps:

(1) Via GEO observations and the adaptive variable-BT segment tracking algorithm, the complex convective organizations of many connected MCSs are decomposed into single MCSs for tracking separately. In this way, anvil clouds are explicitly associated with unique MCSs, which provides the foundation to investigate the process of convective anvil outflow in complex organizations.

(2) Via matching the CCCM data product, the corresponding cloud and radiance vertical structures for these tracked MCSs are retrieved.

In this way, the advantage of the GEO for observing cloud organization and tracking and the advantage of A-Train Constellation for detecting cloud and radiance vertical structures are merged in GATM convective cloud data product. The aim of the GATM is to resolve cloud and radiance structures of convective anvil outflow in complex organizations under a Lagrangian view.

Based on the GATM convective cloud product, the processes of the interaction between radiation and convective anvil outflow at the diurnal time scale are summarized in Figure 8 and described as follows:

(1) **The diurnal variation of convective anvil outflow is primarily driven by the SW radiative heating to clouds.** The results show that daytime SW radiative heating destabilizes the MCS top to enhance the convective anvil outflow. Nighttime LW radiative cooling stabilizes the MCS top to reduce the convective anvil outflow. According to the anvil-radiation mechanisms proposed in previous studies (Lilly, 1988; Hartmann et al., 2018), this is well consistent with the lapse-rate mechanism (the vertical heating structure determines the anvil outflow). Additionally, for the differential radiation mechanism (the horizontal heating difference determines the anvil outflow), it gives a circulation perspective between convective region and its surrounding clear sky to understand the anvil outflow, and suggests that larger clear-sky convergence can promote stronger convection (Gray and Jacobson, 1977; Nicholls, 2015; Wall et al., 2020). The results show that the clear-sky convergence at 01:30 LT (13:30 LT) is stronger (weaker) than the divergence driven by the radiative destabilization. This might imply that the differential radiation mechanism might be more (less) important for the nighttime (daytime) MCS anvil production than the lapse-rate mechanism. At the diurnal time scale, the divergence caused by the SW radiative heating to clouds at 13:30 LT is much stronger than the divergence driven by the clear-sky radiative cooling through circulation at 01:30 LT. As a result, daytime MCSs can produce much stronger anvil clouds than nighttime MCSs (as illustrated in Figure 8), which is well consistent with the observed diurnal variation in the anvil clouds produced by MCSs.

(2) **The radiative budget is modulated by the diurnal variation of convective anvil outflow.** Strong daytime anvil outflow and weak nighttime anvil outflow make the anvil be more distributed at the daytime to reflect more insolation but less distributed at the nighttime to trap less outgoing LW radiation (as illustrated in Figure 8). On average, a secondary radiation cancellation between the net cooling and the net warming occurs on the diurnal time scale, with the cancellation ratio of 27%. This cancellation ratio is sensitive to the diurnal variation of convective outflow but has been rarely studied. According to a simple linear model, the result suggests that the sensitivity of the radiative energy budget to the diurnal-cycle phase shift is approximately -1 W m$^{-2}$ hr$^{-1}$ when the phase shift is in the range between -4 and 8 hr (otherwise the sensitivity has the same magnitude but positive) if the diurnal-cycle amplitude is invariant. And this radiative sensitivity to the phase shift would be proportional to the diurnal-cycle amplitude amplification ratio.

This work explains the interaction processes between radiation and convective anvil outflow at the diurnal time scale. Here, only the association of the radiative energy budget with the anvil area is discussed. But, notably, other anvil properties (e.g., cloud top height and anvil structure) and the evolution for different parts of anvil lifecycles are also important for the radiative energy budget and worthy for investigating in the future work. This work suggests that the SW radiative heating to clouds can explain the diurnal variation of anvil outflow, but it does not guarantee that the diurnal variation of convective anvil outflow is invariant as the climate warms. The sensitivity of the radiative energy budget to the diurnal variation of the convective anvil outflow is assessed in this work. The response of the diurnal variation of the anvil outflow to the climate could be a large component in cloud-climate feedback, but has not been well studied until now. There is a need to investigate the sensitivity of the diurnal variation to the environmental changes in the future work to advance understandings on the cloud-radiation-climate feedback process.

**Acknowledgment**

This work was supported by the NSFC-41875004 and the National Key R&D Program of China (2016YFC0202000). This work was supported by the Jiangsu Collaborative Innovation Center for Climate Change. ZW acknowledges that the GEO images were obtained from the NASA Langley Cloud and Radiation Research Group, http://www-angler.larc.nasa.gov (last access: 10 December 2024).

**Financial support**

This research has been supported by the National Natural Science Foundation of China (grant no. 41875004) and the National Key Research and Development Program of China (grant no. 2016YFC0202000).

**Author contribution**

ZW prepared the original manuscript.

**Data and code availability**

All data used in this study are available online. The GEO images are obtained from the National Aeronautics and Space Administration (NASA) Langley Research Center Atmospheric Science Data Center (https://doi.org/10.5067/MTS01/CERES, NASA/LARC/SD/ASDC, 2017). The GPM is obtained from the Goddard Earth Sciences Data and Information Services Center (GES DISC) at https://doi.org/10.5067/GPM/IMERG/3BHH/07 (Huffman et al., 2023). The CCCM data product are obtained from the National Aeronautics and Space Administration (NASA) Langley Research Center Atmospheric Science Data Center (https://search.earthdata.nasa.gov/). The code is available upon request.

**Competing interests**

The author declares that he has no conflict of interest.

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
