# Peer review of "Anvil-radiation diurnal interaction: Shortwave radiative-heating destabilization driving the diurnal variation of convective anvil outflow and its modulation on the radiative cancellation"

_EGUsphere, 2025_

## Author Response (AR1)

**Response to Anonymous Referee #1**

*Referee #1: This study presents a novel convective cloud data product called GATM, which distinguishes individual MCSs from complex organizations and combine GEOs satellite image with A-Train satellites, making it possible to investigate the anvil production of MCSs and its association mechanisms. Overall, I find the manuscript well written and clearly structured, and I believe it will be a valuable contribution to the literature. I only have minor comments as listed below.*

==Response:== I thank the anonymous reviewer's efforts for reviewing the manuscript. I am very grateful for his/her insightful and helpful comments to improve the clarity and representation of the results. I have carefully taken these comments into account to revise the manuscript accordingly.

*Comments:*

- *When combining GEO satellite images and A-Train satellites, how would an MCS be counted if the A-Train satellite orbit and the center of the MCS are offset (e.g., the MCS1 scenario in Figure 1)? Are there any special procedures in place to address this situation, and would that introduce any uncertainty to the product?*

  ==Response:== For the MCS1, the A-Train-observed cross section consists of multiple profiles. These profiles are all counted to be the MCS1-related profiles. For each profile, its organization and lifecycle information are derived from the GEO-based MCS tracking.

  Specifically, the procedures of constructing the GATM are as follows:

[revised manuscript text omitted]

These descriptions have been added in the revised manuscript.

- *I like how the author discusses the two mechanisms with simple yet clear analyses, but I think the conclusions regarding the two mechanisms and their impact on diurnal cycle of MCSs could be slightly revised for improved clarity*

*(e.g., L304-308). My takeaway is that both mechanisms contribute to anvil production of MCSs. Daytime MCSs are primarily dominated by the lapse rate mechanism with a small contribution from differential radiation mechanism, whereas nighttime MCSs are largely driven by clear-sky convergence with almost no contribution from vertical destabilization. Thus, daytime MCSs produce much stronger anvil clouds than nighttime MCSs, leading to the diurnal cycle.*

Response: Many thanks for your very helpful insights. The conclusions at L304-308 have been revised as: "Overall, the lapse-rate and differential radiation mechanisms both can contribute to the MCS anvil production. In Figure 5e, at 13:30 LT, the MCS divergence directly driven by the radiative destabilization (the rapid decline of the net radiative heating with height) is much stronger than the clear-sky convergence (the differential radiation mechanism). This might imply that daytime MCS anvil production is primarily dominated by the lapse-rate mechanism with a small contribution from the differential radiation mechanism. In Figure 5f, at 01:30 LT, the MCS divergence driven by the radiative destabilization (the increase of the net radiative cooling with height) is smaller than the clear-sky convergence. This might imply that the nighttime MCS anvil production is largely driven by the clear-sky convergence with a small contribution from the vertical radiative destabilization. On the diurnal time scale, the divergence determined by the radiative destabilization at 13:30 LT is much stronger than the divergence driven by the clear-sky radiative cooling through the circulation at 01:30 LT. As a result, daytime MCSs produce much stronger anvil clouds than nighttime MCSs, which leads to the diurnal variation shown in Figure 3.".

- *Figure 8: The conclusion presented in the diagram is unclear based on the accompanying text. While there is more anvil clouds in daytime MCSs compared to night-time MCSs, the existence of night-time MCSs still trap LW radiation and reduce outgoing LW radiation (compared to a clear-sky scenario), thereby contributing to a net warming effects on the Earth. How does "less anvil" lead to "more outgoing LW radiation"? Is the author suggesting less anvil and more outgoing LW radiation over time?*

  Response: It means that less anvil leads to less trapped outgoing LW radiation. Figure 8 has been modified as shown below.

[Figure]

Figure 8. Illustrations of the anvil-radiation diurnal interaction processes.

*Minor comments:*

- *Fig 2c: If Figure 2c shows the accumulated anvil production of MCSs over the development/decay periods, should the unit be km^2 instead?*

**Response:** The unit of the anvil area in each BT image is km^2. The time gap of the GEO BT image is 1 hour. Here, the accumulation of anvil areas for the tracked MCS durations is computed as:

$Accumulated\ anvil\ area = \sum_{i=1}^{D}(A_i \times \delta t)$,   (6)

where $A$ represents the anvil area associated with the tracked MCS in each GEO image, and the subscript "$i$" represents the i-th image for the MCS duration. $D$ represents the MCS duration. $\delta t$ corresponds to the observational time interval and in this work is 1 hour. As a result, the accumulation of anvil production over a period of time has the unit of km^2*hour.

This has been clarified in the revised manuscript.

- *L229 "Here, the hourly anvil producing efficiency refers to the hourly anvil area produced in the MCS decay process.": Since the unit is in %, shouldn't it be further divided by the mean value instead?*

**Response:** Yes, it is further divided by the mean value for quantifying their relative variation on the diurnal time scale.

It has been further clarified in the revised manuscript as: "For the accumulated anvil area produced by MCSs, the hourly mean anvil production and duration are two major contributing factors for its diurnal variation. Here, the hourly mean

anvil production is defined as the accumulated anvil area divided by the lifetime during the decay stage, which represents the hourly mean anvil area produced in the MCS decay process. In Figure 3, by constraining the cold-core-peak BT, the diurnal variation of the accumulated anvil area, hourly mean anvil production and duration are shown. Here, for the same convective peak BT, the diurnal variation is quantified by the diurnal anomalies at different local times divided by the mean value.".

- *How does the clear-sky region define when calculating clear-sky radiative cooling?*

**Response:** The clear sky refers to no clouds above 7 km based on the CCCM data.

This has been clarified in the revised manuscript.

- *L317 "imposes a strong forcing on the Earth radiative energy budget": Forcing usually refers to external factors acting on the system, such as increasing carbon dioxide concentration, aerosols, etc. Recommend changing the term to something like "plays an important role in Earth's energy budget' or something similar.*

**Response:** It has been revised as: "plays an important role in Earth's radiative budget"

- *L376: It seems that the unit "W m-2 K-1" is incorrect as it's inconsistent with the conclusion (L426, W m-2). In addition, it is somewhat unclear how this value should be compared with the high-cloud altitude and tropical anvil cloud area feedbacks from Sherwood et al (2020) and how the difference should be interpreted. This seems to be an apple-to-orange comparison as the unit differ, and the feedback values from Sherwood et al (2020) should already be weighted by the ratio of regional area to global area.*

**Response:** This sentence has been revised as: "Although the diurnal-cycle climate feedback and its relevant mechanism are still missing in current climate studies, cloud-resolving models and observations both suggested that the diurnal cycle of cloud coverage could have a positive phase shift with surface temperature rising (Yin and Porporato, 2019; Gasparini et al., 2021; Wang et al., 2022). Nevertheless, recent studies have suggested that the diurnal-cycle features of clouds (e.g., diurnal-cycle amplitude and phase) simulated by climate models have significant biases compared against that in the observation (Yin and Porporato, 2017; Chen et al., 2022; Zhao et al., 2023). As a result, predicting the diurnal-cycle climate feedback remains challenging but might be important for understanding the climate sensitivity. Here, diurnal-cycle climate feedback can be largely determined by multiplying the response of the diurnal cycle to the surface temperature by the sensitivity of the radiative budget to the changes in the diurnal cycle. The response of the diurnal cycle is still an uncertain aspect in

the future climate, whereas the radiative sensitivity to the diurnal cycle can be assessed."

- *L381: It would be helpful to provide an estimate of the phase shift in response to warming based on previous literatures, such as an order of magnitude or an estimated range.*

**Response:** Thanks for your suggestion. The sensitivity of NetCREs to the diurnal-cycle phase of convective anvil outflow is approximately -1 W m$^{-2}$ hr$^{-1}$ when the phase shift is in the range between -4 and 8 hr (otherwise the sensitivity has the same magnitude but positive). Notably, the radiative sensitivity to the diurnal-cycle phase is proportional to the diurnal-cycle amplitude amplification ratio ($\lambda$) in Figure 7b, with the regression coefficient of approximately 1. As the diurnal-cycle amplitude is stronger with the amplification ratio $\lambda$, the radiative sensitivity to the phase shift would be amplified by multiplying by $\lambda$. As a result, if the climate response of the diurnal cycle of the convective anvil outflow to the temperature can be known, the sensitivity that is assessed here might be useful for inferring to its feedback strength.

*Text:*

- *L107: should be "and" rather than comma (i.e., cold cores and cold centers).*

**Response:** It has been corrected as: "i.e., cold cores and cold centers"

- *L159: Jule -> July*

**Response:** It has been corrected as: "July".

- *L379: "The" should be lowercase*

**Response:** It has been corrected as: "the".

**Response to Anonymous Referee #2**

*Referee #2: Review of "Anvil-radiation diurnal interaction: Shortwave radiative-heating destabilization driving the diurnal variation of convective anvil outflow and its modulation on the radiative cancellation"*

*In this manuscript, the author presents novel research into the lifecycle of deep convective clouds, and how their anvil cloud amounts cloud radiative effect vary with the diurnal cycle. This study builds on a relatively unexplored area of the dependence of anvil cloud radiative effect on the diurnal cycle but unlike previous studies (e.g. Nowicki & Merchant, 2004; Bouniol et al., 2021; Jones et al., 2024) focuses on the impact of diurnal differences in anvil cloud processes rather than shifts in the timing of convection. This research also expands on previous studies into how the diurnal cycle of radiative heating affects anvil development (e.g. Wall et al. 2020; Gasparini et al. 2022), provides additional evidence for these processes by placing observations of anvils within the diurnal cycle and considering how changes in these processes may result in climate feedbacks. To conduct this research, the author has developed a novel dataset that combines Lagrangian properties of mesoscale convective systems tracked using geostationary satellite images with collocated retrievals from Cloudsat, MODIS and CERES. This new dataset helps resolve some of the shortcomings with studies using only one source of observations, and may also provide further results in future studies.*

*The manuscript is generally well written and presented, and presents impactful results. The introduction provides a clear background to the problem, although could include a little more discussion of some more recent literature. The methods and results are concisely described, and the discussion and conclusions are well described.*

==**Response:**== Thanks for your time and efforts for reviewing this paper. I am sincerely grateful for your insightful and thorough comments for improving the clarity and understanding of this paper. I have taken all comments into account and revised the manuscript carefully.

*Overall, I have three main criticisms of the manuscript:*

1. *Use of non-standard acronyms and terminology makes some parts of the manuscript confusing, and in addition parts of the methods and results sections could be reworded to improve clarity*

==**Response:**== The non-standard terminologies have been corrected:

(1) "convective peaking strength" has been replaced with "convective peak BT" (see the response to *Line 180*);
(2) "anvil production" has been replaced with "lifetime-accumulated anvil area" (see the responses to *Line 192* and *Line 229*);

(3) "radiative energy forcing" has been replaced with "radiative energy contribution" (see the responses to *Section 4* and *Line 329*).

To avoid confusion, the relevant equations for computing them have been provided and explained in the revised manuscript (see the following responses).

In addition, methods and results have been carefully revised and more descriptions have been provided for improving clarity.

2. *There is a lack of discussion of sources of bias in the data and how this may effect the results, and some of the decisions made in designing the study, while valid, would be improved by adding a description explaining why these choices were made.*

==Response:== The sources of bias in the data and its limitations for analyses have been clarified in the revised manuscript (see responses to *Line 143*, *Line 159* and *Section 4*).

More descriptions have been added for explaining the design of this study (see responses to *Line 130, Line 180* and *Line 181*).

Additionally, for the decisions made in identifying the non-precipitating anvil, although the BT threshold of 260 K is useful for identifying tropical high clouds (Minnis et al., 2008; Minnis et al., 2011), much the area of detrained thin cirrus of the BT warmer than 260 K is not well identified (Gasparini et al., 2022; Sokol and Hartmann, 2020; Berry and Mace, 2014). It has been demonstrated that 95% of deep convective clouds and as much of the anvil cloud as possible can be identified with the least contamination from lower-level clouds by using the threshold of 260 K (Yuan and Houze, 2010; Yuan et al., 2011; Chen and Houze, 1997). This has been clarified in the revised manuscript.

3. *The results in section 3.2 do not consider the different parts of the anvil lifecycle as discussed in section 3.1, although deeper discussion of changes in anvil CRE throughout the lifecycle may be more suitable for a subsequent paper.*

==Response:== Different anvil lifecycles are not discussed in section 3.2 and this topic is interesting and worthy for investigating in the future work, especially for showing the evolution in microphysics, anvil structure and CRE for different parts of anvil lifecycles.

It has been clarified in the discussion of the revised manuscript: "notably, other anvil properties (e.g., cloud top height and anvil structure) and the evolution for different parts of anvil lifecycles are also important for the radiative energy budget and worthy for investigating in the future work.".

*Comments:*

*Line 31: Referencing some of the recent evidence for alternative anvil CRE feedback mechanisms (e.g. Sokol et al., 2024; McKinnon et al. 2024; Raghuraman et al. 2024) could help reinforcement this point about novel feedback processes and the importance of this study*

**Response:** Thanks for providing these references. They have been added in the revised manuscript.

*Line 44: The anvil cloud area feedback is commonly explained through the radiative Iris effect (Bony et al., 2016), although this is somewhat disputed. More recent research has highlighted the importance of anvil structure to anvil CRE (see previous comment for references)*

**Response:** More explanations according to recent studies have been discussed here: "As the climate warms, the changes in the atmospheric state can influence the properties of the anvil produced by tropical convection. The variations in the anvil area and the proportion of thin clouds relative to thick clouds both can alter the radiative cancellation. Bony et al. (2016) suggested that the enhanced upper-tropospheric stability can reduce the convective outflow and anvil cloud fraction. As a result, if the anvil opacity stays the same, this reduction of anvil areas is expected to weaken the radiative cancellation to impose negative feedback on the climate. However, although the tropical high cloud area has a reduction as the climate warms, Sokol et al. (2024) suggested that the high cloud opacity is not fixed but the reduction of thick cloud area is stronger compared with thin clouds. This opacity climate response leads to a higher proportion of thin clouds relative to thick clouds and thus results in a positive climate feedback process (Sokol et al., 2024).".

*Line 59: Suggestion "in the widely-used Euler-view grid data set" "in widely-used Eulerian gridded data"*

**Response:** It has been modified as: "in widely-used Eulerian gridded data".

*Line 130: What happens to the regions between the definitions of cold-core and anvil with precip > 1mm/hour and < 6 mm/hour?*

**Response:** The regions of the precipitation rate more than 1 mm/hour are defined as the precipitating region. The threshold of 1 mm/hour is used to distinguish precipitating/non-precipitating region, which is consistent with the definition in Yuan and Houze (2010). Another threshold of 6 mm/hour is used to distinguish whether the cloud system has heavy precipitation. If it has the heavy precipitation, it is identified as the mesoscale convective system, which is also consistent with the definition in Yuan and Houze (2010). As a result, the region of the precipitation rate of 1~6 mm/hour is the precipitating region but not the heavy precipitating region.

To improve the clarity, this paragraph has been rewritten as: "In the tropics, light precipitation (<1 mm/hour) is difficult to be accurately identified by the GEO-based precipitation estimate (Tian et al., 2009) and contributes to only 9%–18% of the total precipitation (Yuan and Houze, 2010). Thus, the threshold of 1 mm/hour is used to distinguish precipitating/non-precipitating region, which is consistent with Yuan and Houze (2010). The MCSs are commonly identified as the tropical deep cloud systems with heavy precipitation (Williams and Houze, 1987; Fu et al., 1990; Yuan and Houze, 2010). Here, the heavy precipitation event is defined as that the area of the precipitation larger than 6 mm/hour exceeds 1000 km^2, which is consistent with the definition in Yuan and Houze (2010). The MCS is defined as the OS of the heavy precipitation, the cold-core-peak BT colder than 220 K, and the duration over 5 hours. These MCSs represent the cold and long-lived OSs in CCOs and contribute to most of tropical precipitation and anvil clouds (Wang and Yuan, 2024).".

*Line 143: Are the ToA fluxes provided by the Cloudsat algorithm (i.e. are 1D slices through observed systems) or are 2D fields provided by CERES (covering the whole anvil)?*

**Response:** The vertical profiles of cloud and radiance are provided by the CALIPSO-CloudSat-CERES-MODIS (CCCM) data set, which were developed in Kato et al. (2011). The 2D field of the full picture of convective organization and its temporal evolution are provided by GEO. The GATM is combine the GEO-based tracking data and the CCCM data.

To improve the clarity of this part, a new Section 2.2 has been added to introduce the CCCM data and the Section 2.3 has been revised to focus on the procedures of constructing the GATM: "Specifically, the procedures of constructing the GATM are as follows:

**(1) Matching the A-Train-observed profiles with GEO image pixels:** According to the longitude and latitude of each profile in the A-Train orbit and the satellite overpass time, the A-Train profile is matched with the nearest GEO image pixel, with the distance no more than 3 km and the observational time difference no more than half an hour. As illustrated in Figure 1, the orbit of the A-Train corresponds to four GEO-observed MCSs (MCS1-3 are connected and MCS4 is isolated). By matching the A-Train orbit with the GEO image pixels, the association of these A-Train profiles with the GEO-observed MCSs is distinguished.

**(2) Deriving the organization and lifecycle information from the GEO for the A-Train-observed profile:** The organization and lifecycle information for the A-Train and GEO matched profiles in the first step is further derived from the GEO-based MCS tracking. As illustrated in Figure 1, for the MCS1, the A-Train-observed cross section consists of multiple MCS1-related profiles. The GEO provides a full picture of the MCS1 spatial organization structure and its life cycle. In this work, for each MCS1-related profile, the information derived from the GEO includes: a) the distance to the cold-core centroid, which corresponds to its relative position in the MCS; b) the coldcore BT and cold-center BT, which represent the MCS organization structure; c) the cold-core-peak BT, its life stages (developing, peaking or decaying) and the time relative to the convective peaking time, which reflect the MCS life cycle. In this way, these A-Train and GEO matched profiles receive a Lagrangian perspective from the GEO-based MCS tracking.

For the GEO-A-Train matching process, the uncertainty might result from the difference in the temporal resolution between the GEO and A-Train satellites. The GEO images have hourly resolution, whereas the A-Train-observed profiles are instantaneous. Although the observational time difference between the GEO and A-Train satellites is constrained no more than half an hour for matching, clouds could vary significantly during this half an hour. The temporal resolution of the new-generation GEO has achieved 2.5 mins (e.g., Himawari-8 launched in 2014) and benefits tracking in recent years (Daniels et al., 2020; Yang et al., 2023), but the best observational period of the A-Train Constellation is during the year 2006-2011 (Kato et al., 2011). Thus, in this work, to take advantages of the A-Train Constellation, the GEOs of the same period during 2006-2011 are used to construct the GATM."

*Line 159: Note that removing systems that intersect the edge of the domain may cause a low bias in the number of larger and longer lived systems included in the statistics, but it is difficult to account for (see deWitt et al. 2024).*

Response: Thanks for your note. This uncertainty has been clarified in the revised manuscript: "Normally, large and long-lived cloud systems are more likely to touch the edge of the domain than small and short-lived systems. Thus, removing the systems that intersect with the edge can result in a low bias in the number of the large and long-lived systems (Dewitt and Garrett, 2024; Dewitt et al., 2024).".

*Figure 2: It may be clearer to split 2a into two plots, one showing the number of MCSs observed with each peak BT, and another showing the % contributions to total anvil coverage from developing and decaying anvils. It would be useful to show lines for the total duration and the anvil production in figures 2b/c*

Response: Figure 2 has been modified as shown below.

[Figure]

Figure 2. Life cycles of MCSs via adaptive variable-BT tracking. (a) The sample number of the tracked MCS life cycles of different cold-core-peak BT in June-August from 2006 to 2011 over the TWP. (b) The relative contribution fractions to total anvil areas for development (the blue line) and decay (the red line) stages, respectively. The average duration (c) and lifetime-accumulated anvil area (d) of different cold-core-peak BT. The blue and red represent the quantities related to the development and decay stages, respectively. The black dash line indicates their differences and the black solid line indicates their sum. The error bars indicate the 95% confidence intervals of the means based on the t test.

*Line 180: It may be clearer to refer to "convective peak BT" rather than "convective peaking strength" throughout. While colder BTs are strongly correlated with more intense convection, they can also be due to meteorological or other differences*

**Response:** "convective peaking strength" has been replaced with "convective peak BT" throughout the paper.

This sentence has been further clarified as: "The BT can be influenced by atmospheric and surface emission and cloud optical properties. Normally, over tropical oceans, the variation of the cold BT (< 230 K) is deemed to be more correlated with the convection activity, whereas the variation of the warm BT could be more influenced by the disturbance in surface emission (Fu et al., 1990; Hendon and Woodberry, 1993). In this work, only the convective peak BT colder than 220 K is considered and the least identified variation of the peak BT is 5 K, which means the

fluctuations less than 5 K (that might be contributed by the variation of atmospheric and surface emission) are filtered out. Thus, smaller convective peak BT values are largely contributed by more intense convection.".

*Line 181: The division of MCS lifecycle is different to previous methods for separating convective lifecycles, such as the growing, mature, dissipating (e.g. Futyan and del Genio, 2007) or splitting developing and decaying based on the time which the anvil reaches its largest horizontal extent (e.g. Roca et al. 2017). The choice of coldest BT is valid, and could be a useful consideration regarding anvil CRE, but I would like to see a discussion on why this is chosen and how it differs from other methods of segmenting the lifecycle. In particular, the time of coldest BT will tend to occur earlier in the lifecycle than the largest area. Also, MCSs which last for multiple days may go through multiple cycles of invigoration with the diurnal cycle, resulting in multiple peaks of the anvil BT throughout the MCSs lifetime. Choosing a single time may lead to anomalies with these systems, although they are likely rare and so should not have significant effects on any of the statistics.*

**Response:** The part in the revised manuscript has been clarified as: "The life cycle is simply separated into two stages: development (before peaking) and decay (peaking and after peaking). The life cycle of convection can be split into more detailed stages, such as the initiation, growth, mature and dissipation (Futyan and Del Genio, 2007; Wall et al., 2018; Yang et al., 2024), or discriminated according to the area variation (Chen and Houze, 1997; Roca et al., 2017). For this work, the convective peak BT is used as a simple useful constraint on the convective intensity to split life cycle stages. Normally, time of the convective peak BT will tend to occur earlier in the life cycle than the largest area (Futyan and Del Genio, 2007; Yang et al., 2024).The decay stage after the time of the convective peak BT (defined in this work) corresponds to the sum of mature and dissipation stages in previous studies (Futyan and Del Genio, 2007; Yang et al., 2024). Wang and Yuan (2024) have validated that the convective peak BT can well constrain the convection precipitating and producing anvil clouds in the life cycle, even though the life cycle is complicated.".

*Line 192: "Anvil production" sounds like a rate, e.g. the increase in anvil area per hour, rather than the total amount. It may be clearer to refer to this quantity as accumulated anvil area or total anvil coverage*

**Response:** Thanks. "anvil production" has been replaced with "accumulated anvil area" in the revised manuscript.

*Line 196: While the colder peak BT anvils produce larger anvils, I'm not sure it can be stated that they are more efficient at producing anvils. I would consider efficiency as a measure of anvil area vs core area/updraft strength*

**Response:** This sentence has been revised as: "the MCSs of colder peak BT values are much more long-lived and are accompanied with stronger anvil outflow".

*Line 229: It is unclear what "hourly anvil producing efficiency" means. Is this just the mean anvil area over the decaying portion of the MCS lifetime? I.e. the red line in fig 2c divided by the lifetime?*

**Response:** Yes, it is the accumulated anvil area divided by the lifetime in the decay stage.

It has been clarified as: "For the accumulated anvil area produced by MCSs, the hourly mean anvil production and duration are two major contributing factors for its diurnal variation. Here, the hourly mean anvil production is defined as the accumulated anvil area divided by the lifetime during the decay stage, which represents the hourly mean anvil area produced in the MCS decay process.".

*Line 231: Would it be simpler to say that there is more variation in the average area of anvils than the anvil lifetime with the diurnal cycle?*

**Response:** Thanks. This sentence has been modified as: "Overall, there is more variation in the average area of anvils than the anvil lifetime with the diurnal cycle.".

*Line 265 onwards (discussion of figure 4): Is the sampling of anvil heating rates uniform across the anvil? Compositing 1D CloudSat overpasses to produce statistics across 2d anvils can lead to biases. For example, if the overpass is directly through the core of the MCS, then a larger proportion of the core will be sampled than the anvil further from the core, leading to a cold bias in the measured fluxes. The preprint of Igel et al. (2014) discussed the statistics of this, but I don't think that a final version was published. A simple way to check this would be to plot the number of CloudSat samples vs the distance to the MCS cold core; if the sampling is uniform then this should show a linear relationship.*

**Response:** The sampling across the anvil is not uniform. There are more samples when the distance is closer to the cold core, whereas at the edge of 300 km away from the cold core there are only dozens of samples. It might be because the large systems are less frequent than small systems.

The method of combining the GEO and A-Train satellites (used in this work, Figure 4) may not lead to a strong bias compared with the method of only using the CloudSat. If only using the CloudSat, the identification of a convective system depends on the whether the convective pillar is observed. In other words, if the overpass is not through the core of the MCS, the anvil cannot be identified. When producing statistics of the 2-D cloud structures by only using the CloudSat, the core of the MCS is composited by more cloudy samples and fewer clear-sky samples, whereas the edge of the MCS far away from the core is the composite of fewer cloudy samples and more clear-sky samples. This might lead to a bias for the anvil structure further from the core, since it is mixed with a large number of clear-sky samples. On the other hand, by combining the GEO, according to the provided full picture of the

convective system, even though the overpass of the A-Train is not through the core of the MCS, the anvil still can be identified (as illustrated in Figure 1). In this way, when producing the statistics of the 2-D cloud structures, none of clear-sky samples is included and thus the anvil heating structure further from the core is not biased by clear-sky samples. This has been clarified in the section 2.3 of the revised manuscript.

*Figure 4: This is a really nice figure. It might be clearer to use different colour scales for destabilisation and divergence to emphasise the differences shown*

**Response:** Thanks. The color scale of the Figure 4 has been modified as follows:

[Figure]

Figure 4. Radiative heating rate, heating destabilization and divergence caused by LW, SW and net radiances for MCSs at 13:30 and 01:30 LT, respectively. (a-c) The LW, SW and net heating rate at 13:30 LT. (d) The LW heating rate at 01:30 LT. (e-h) and (i-l) similar to (a-d) but for heating destabilization and divergence. The black contours represent cloud fraction.

*Line 305: This discussion could be enhanced by also including the average column heating rates, destabilisation and divergence for anvils in figure 5, along with the clear-sky rates, so that the differences can be seen more clearly*

**Response:** Figure 5 has been modified as shown below.

[Figure]

Figure 5. The composited profiles of radiative heating rate, heating destabilization and divergence at 13:30 and 01:30 LT, respectively. (a) The LW, SW and net heating rate at 13:30 LT. (b) The LW heating rate at 01:30 LT. (c-d) and (e-f) similar to (a-b) but for heating destabilization and divergence, respectively. The red, blue and black solid lines represent quantities caused by LW, SW and net radiances for clear sky, respectively. The black dashed line represents the quantity caused by the net radiance for MCSs, whose details are shown in Figure 4.

*Section 4: How does the interaction between the MCS lifecycle and the timing of the CloudSat overpass affect the measured anvil CRE? E.g. for a Cloudsat overpass at 13:30 LT, an MCS than peaks at 12:00 LT will be in the decaying phase while one that peaks at 15:00 LT will be observed in the developing phase. How much does the difference in lifecycle (and possible difference in anvil area/structure) affect the measured CRE compared to the change seen over the diurnal cycle. This contribution may be beyond the scope of this manuscript to explore in depth, but should be mentioned in the text regardless.*

**Response:** The analyses of section 4 depend simply on the GEO-based MCS tracking data set (introduced Section 2.1) and the A-Train observation is not used in this section. The CREs are computed based on the GEO observed albedo and OLR. Thus, the CREs during the whole life of the MCS are available and the first concern about the interaction between the MCS lifecycle and the timing of the CloudSat overpass can be largely eliminated.

The method of computing CREs has been clarified in the Section 2.1 in the revised manuscript as: "For the all-sky radiative flux at the top of atmosphere (TOA), the hourly broadband shortwave albedo ($\beta$) and outgoing longwave radiative flux (LWRF) images of 0.05° resolution are derived from the GEO radiometers. For the clear-sky radiative flux at the TOA, the hourly insolation and the broadband clear-sky LWRF and reflected shortwave radiative flux (SWRF) at each grid of 0.05° are allocated from the Clouds and the Earth's Radiant Energy System (CERES) synoptic 1-degree (SYN1deg) product (Doelling et al., 2016). The CREs are defined as the difference of the predicted TOA clear-sky upwelling radiative flux relative to the observed TOA all-sky radiative flux:

$$LW\ CRE = LWRF_{clr} - LWRF_{obs}, (1)$$

$$SW\ CRE = SWRF_{clr} - \beta_{obs} \times Insolation_{obs}, (2)$$

$$Net\ CRE = LWCRE + SWCRE, (3)$$

where the subscripts "clr" and "obs" represent the clear sky and the observed all sky, respectively. The CRE represents the TOA radiative energy budget altered by clouds per square meter and per hour. Thus, the impact of the MCS on the radiative energy budget depends largely on the CRE, area and duration. For example, short-lived and small MCSs may have strong CREs but contribute to only a limited energy disturbance, since they only impact a small region during a short time. The CREs of long-lived and large MCSs may not be strong but the radiative energy budget can be strongly altered by them, since they are long-lasting to impact a large region. As a result, to evaluate the impact of the MCS on the radiative energy budget, the radiative energy contribution (REC) is defined as sum of CRE for non-precipitating anvils over their entire area and lifetime:

$$REC = \sum_{i=1}^{D} \sum_{j=1}^{N} CRE_{i,j} \times \delta area_{i,j} \times \delta t. (4)$$

Here, $\delta area$ is the observational grid area of 0.05° resolution, which is a function of latitude. $\delta t$ is the observational time interval, which is 1 hour in this work. The subscript "$i$" represents the $i$-th time and $D$ is the duration of the MCS. The subscript "$j$" represents the $j$-th grid of the non-precipitating anvil and $N$ is the total number of grids covered by non-precipitating anvil clouds."

For the second concern, the CRE and REC can be affected by many factors. It has been clarified in the revised manuscript as: "Although the radiative energy budget is sensitive to many anvil properties (e.g., the top temperature, structure and life cycle), only the association of the radiative energy budget with anvil area coverage is focused in this section. And for the LW and SW radiative energy budget shown in Figure 6b-c, the LW and SW RECs are closely correlated with the anvil contribution

shown in Figure 6a, with the pattern correlations of 0.97 and -0.8 at the 99% significant level, respectively.".

*Figure 6: Why is this shown in units of kJ, rather than W m-2? Is this the sum of CRE for anvils over their entire area and lifetime? Could the area weighted ToA flux be used instead to show the CRE in than W m-2? It may help the explanation of the secondary cancellation to show all of these components. i.e. show the total/average anvil area and lifetime for MCSs peaking at different local times*

==Response:== The CRE represents the TOA radiative energy budget altered by clouds per square meter and per hour. Thus, the impact of the MCS on the radiative energy budget depends largely on the CRE, area and duration. For example, short-lived and small MCSs may have strong CREs but contribute to only a limited energy disturbance, since they only impact a small region during a short time. The CREs of long-lived and large MCSs may not be strong but the radiative energy budget can be strongly altered by them, since they are long-lasting to impact a large region.

Yes, the radiative energy contribution (REC) is defined as sum of CRE for non-precipitating anvils over their entire area and lifetime (see the response to the last comment).

The diurnal variation of the MCSs producing anvil clouds has been presented in Figure 3. In the revised Figure 6 (shown below), the anvil area contributed by MCSs of different peak BTs and LTs has been added. The anvil contribution fraction refers to the fraction of the non-precipitating anvil area produced by MCSs of different peak BTs and LTs relative to the anvil area produced by all observed MCSs.

[Figure]

Figure 6. Anvil contribution and radiative cancellation on the diurnal time scale. (a) Anvil contribution fraction of MCSs of different peak BT and times. (b-d) The LW, SW and Net REC caused by MCSs of different peak BT and times.

*Line 329: In generally used terminology this is a radiative effect, not a radiative forcing. It may be clearer to change REF to something like "radiative energy contribution" or discuss the area weighted CRE as discussed in the previous comment*

**Response:** Thanks. It has been modified as "radiative energy contribution (REC)" (see the response to previous comments).

*Line 339: Are these factors important for the secondary cancellation? Anvil temperature is important for LW cooling, and is important in the primary cancellation. The anvil area coverage is more important for considering feedbacks in anvil area. It might also be useful to note the importance of anvil structure in the radiative cancellation (e.g. Berry and Mace, 2014), which has been shown to be increasingly important for the anvil radiative feedbacks in recent years (e.g. Raghuraman et al. 2024, Sokol et al. 2024, McKim et al. 2024)*

**Response:** Yes, they are important and the observed cancellation in Figure 6 should be attributed to many factors, but here only its association with the diurnal variation of convective anvil outflow is focused.

This has been clarified in the revised manuscript: "The observed radiation cancellation in Figure 6 might result from many factors. The anvil-top temperature is important for outgoing LW radiation and accounts for the primary radiative cancellation (Kiehl, 1994). Additionally, the negative CREs caused by thick anvil clouds can be partially balanced by the positive CREs of optically thin cirrus clouds and thereby the anvil structure (i.e., the ratio of thin cirrus clouds relative to thick clouds) is also important for the radiative cancellation (Berry and Mace, 2014). Recent studies suggested that the anvil structure is an important determinant of the anvil radiative climate feedbacks (Mckim et al., 2024; Raghuraman et al., 2024; Sokol et al., 2024). Moreover, the diurnal variation of anvil clouds produced by MCSs also can affect the radiative cancellation, particularly for the secondary radiation cancellation at the diurnal time scale. For example, if the diurnal cycle of the MCS anvil contribution (shown in Figure 6a over oceans only) has a positive phase shift, the anvil cloud budget would be redistributed at the diurnal time scale, with relatively more daytime anvil clouds and less nighttime anvil clouds. As a result, the diurnal-cycle amplitude of the SW RECs can be further amplified to reduce the secondary net radiation cancellation ratio and increase net cooling effects. Similarly, the diurnal variations of the anvil-top temperature and anvil structure produced by MCSs are also important for the secondary net radiation cancellation."

*Line 340: This also depends on the present day diurnal cycle of DCCs, which is very different between land and ocean.*

**Response:** It has been clarified as: "over oceans only".

*Figure 7: The colour scales could be adjusted to show differences more clearly*

**Response:** The color scale of Figure 7 has been modified as shown below.

[Figure]

Figure 7. Sensitivity of the net radiation cancellation to diurnal variations of the MCS producing the anvil. (a-b) The sensitivity of the secondary net radiation cancellation ratio and net CREs, respectively, to distinct climatology of diurnal variation of MCS anvil production. The red dot represents the current climatology of the diurnal cycle.

*Line 376: Should the units in the value of 11 W m-2 K-1 be W m-2 instead? It is difficult to compare this value to the values from Sherwood et al., as it is not the average global response to a temperature change. How much do you expect the diurnal cycle of anvils to respond to temperature change? Interestingly the value of 11 W m-2 is similar to the value estimated by Nowick and Merchant (2004) for a 1 hour shift in timing of convection over land.*

==Response:== The previous description has been replaced with: "the sensitivity of NetCREs to the diurnal-cycle phase of convective anvil outflow is approximately -1 W m$^{-2}$ hr$^{-1}$ when the phase shift is in the range between -4 and 8 hr (otherwise the sensitivity has the same magnitude but positive). Notably, the radiative sensitivity to the diurnal-cycle phase is proportional to the diurnal-cycle amplitude amplification ratio ($\lambda$) in Figure 7b, with the regression coefficient of approximately 1. As the diurnal-cycle amplitude is stronger with the amplification ratio $\lambda$, the radiative sensitivity to the phase shift would be amplified by multiplying by $\lambda$. As a result, if the climate response of the diurnal cycle of the convective anvil outflow to the temperature can be known, the sensitivity that is assessed here might be useful for inferring to its feedback strength.".

*Technical corrections:*

*Line 24: These two sentences might be in the wrong order*

**Response:** It has been modified as: "Tropical convective regions are usually characterized by abundant convective activities and anvil clouds (Houze, 2004; Yuan and Houze, 2010). Anvil clouds have strong interactions with radiation.".

*Line 206: "decay process" -> "decay period"*

**Response:** It has been corrected as: "decay period".

*Line 219: "The MCSs of the peak BT at 195 K have 5310 samples and warmer MCSs are more" rephrase for grammar/clarity*

**Response:** It has been rephrased as: "The MCSs in each bin of the peak BT from 195-220 K have thousands of samples for investigating their diurnal variations and are the major source of anvil clouds.".

*Line 244/246: "thus to increase" -> "thus increasing"*

**Response:** It has been corrected as: "thus increasing".

*Line 322: Unfinished sentence*

**Response:** It has been corrected as: "the sensitivity of the radiative energy budget to the diurnal variation in the cloud water budget is investigated".

*Line 342: "re-disturbed" -> "redistributed"*

**Response:** It has been corrected as: "redistributed".

[revised manuscript text omitted]